# Retaining Suboptimal Actions to Follow Shifting Optima in Multi-Agent Reinforcement Learning

**Yonghyeon Jo, Sunwoo Lee, Seungyul Han**[*]
Graduate School of Artificial Intelligence
Ulsan National Institute of Science and Technology (UNIST)
Ulsan, South Korea 44919
{yonghyeonjo,sunwoolee,syhan}@unist.ac.kr

## Abstract

Value decomposition is a core approach for cooperative multi-agent reinforcement learning (MARL). However, existing methods still rely on a single optimal action and struggle to adapt when the underlying value function shifts during training, often converging to suboptimal policies. To address this limitation, we propose Successive Sub-value Q-learning (S2Q), which learns multiple sub-value functions to retain alternative high-value actions. Incorporating these sub-value functions into a Softmax-based behavior policy, S2Q encourages persistent exploration and enables $Q^{\text{tot}}$ to adjust quickly to the changing optima. Experiments on challenging MARL benchmarks confirm that S2Q consistently outperforms various MARL algorithms, demonstrating improved adaptability and overall performance. Our code is available at https://github.com/hyeon1996/S2Q.

## 1 Introduction

Multi-agent reinforcement learning (MARL) has emerged as a powerful and versatile framework for solving complex sequential decision-making problems involving multiple interacting agents (Canese et al., 2021; Li et al., 2022; Wen et al., 2022). Its applicability spans a wide range of domains, including robotic control, autonomous driving and traffic management, intelligent manufacturing, as well as strategic and competitive environments such as real-time strategy games (Canese et al., 2021; Li et al., 2022; Orr & Dutta, 2023; Shalev-Shwartz et al., 2016; Bahrpeyma & Reichelt, 2022; Vinyals et al., 2019). Within the centralized training with decentralized execution (CTDE) paradigm, where agents can leverage centralized information during training while acting independently at execution time, value-decomposition-based credit assignment methods (Sunehag et al., 2017; Rashid et al., 2020a; Wang et al., 2020a) have proven highly effective across many MARL benchmarks.

Nevertheless, conventional approaches face fundamental challenges due to the requirements of decentralized execution. In particular, QMIX (Rashid et al., 2020a), one of the most representative CTDE methods, enforces the Individual-Global-Max (IGM) condition (Son et al., 2019) through a monotonicity constraint, ensuring that the joint action-value function increases monotonically with respect to the utility of each agent. Although this design enables competitive performance across diverse tasks, it also restricts the expressiveness of the joint action-value function, limiting its ability to represent complex interactions among agents. Several lines of work have attempted to overcome this limitation. For instance, Rashid et al. (2020b) introduces an unconstrained target network to improve the fidelity of optimal-action estimation, while Zhou et al. (2022) incorporates auxiliary information to mitigate representational bottlenecks under partial observability. More recent studies reformulate the value function into forms that are more amenable to decomposition (Shen et al., 2022; Li et al., 2024), further improving stability and performance.

Despite these advances, existing methods still struggle when the optimality of the value function shifts during training. This challenge arises not only from the representational rigidity imposed by the monotonicity constraint but also from the heavy reliance on $\epsilon$-greedy policy in the typically

---

[*]Correspondence to: Seungyul Han.

large joint action spaces of MARL. To address these issues, we propose Successive Sub-value Q-learning (S2Q), a novel MARL framework that successively learns multiple sub-value functions, each designed to identify a distinct suboptimal action. Leveraging these sub-value functions, S2Q replaces naive $\epsilon$-greedy policy with a Softmax-based strategy that encourages persistent visitation around promising joint actions. This allows S2Q to accurately detect shifts in the optimal action and rapidly adapt its policy, thereby avoiding premature convergence to suboptimal solutions. We further provide theoretical and empirical analyses to validate the effectiveness of our approach, demonstrating that S2Q achieves substantial improvements over other recent MARL methods on challenging benchmarks, including the StarCraft II Multi-Agent Challenge (Samvelyan et al., 2019) and Google Research Football (Kurach et al., 2020).

## 2 PRELIMINARIES

### 2.1 DECENTRALIZED POMDPS

In cooperative multi-agent RL, the problem is formulated as a Decentralized Partially Observable Markov Decision Process (Dec-POMDP), represented by the tuple $G = \langle \mathcal{S}, \mathcal{A}, P, r, O, \mathcal{O}, N, \gamma \rangle$, where $\mathcal{S}$ denotes the state space, $\mathcal{A}$ the action space of each agent, $P$ the transition probability function, $r$ the reward function, $O$ the observation function, $\mathcal{O}$ the observation space, $N$ the number of agents, and $\gamma$ the discount factor(Oliehoek et al., 2016). At each timestep $t$, the environment resides in a global state $s_t \in \mathcal{S}$, while agent $i$ receives a local observation $o_t^i := O(s_t) \in \mathcal{O}$. Based on its trajectory history $\tau_t^i := (o_0^i, a_0^i, \ldots, o_t^i)$, agent $i$ selects an action $a_t^i \in \mathcal{A}$ according to a decentralized policy $\pi^i$. After receiving the joint action $\mathbf{a}_t = (a_t^1, \ldots, a_t^N)$ from all agents, the environment transitions to the next state $s_{t+1} \sim P(\cdot \mid s_t, \mathbf{a}_t)$ and the reward $r_t := r(s_t, \mathbf{a}_t)$. The overall objective of MARL is to maximize the expected discounted return, $\mathbb{E}[\sum_{t=0}^{\infty} \gamma^t r_t]$.

### 2.2 VALUE DECOMPOSITION UNDER CTDE

Under the Centralized Training with Decentralized Execution (CTDE) paradigm, agents act independently during execution while utilizing global state information during training. Among the most prominent methods in this setting is QMIX(Rashid et al., 2020a), which employs a monotonic mixing function to enforce the Individual-Global-Max (IGM) condition, ensuring that maximizing individual utility $Q^i$ cannot reduce the joint value $Q^{\text{tot}}$. Although such alignment is beneficial, the monotonicity constraint limits the expressiveness of $Q^{\text{tot}}$, often hindering effective minimization of the temporal-difference (TD) loss. To address this limitation, Weighted QMIX (WQMIX)(Rashid et al., 2020b) introduces an auxiliary unconstrained joint action-value function $Q^*$:

$$\mathbb{E}_{(s_t, \boldsymbol{\tau}_t, \mathbf{a}_t) \sim \mathcal{B}} \left[ (Q^*(s_t, \boldsymbol{\tau}_t, \mathbf{a}_t) - y_t)^2 \right], \quad y_t = r_t + \gamma Q^*_{\text{targ}}(s_{t+1}, \boldsymbol{\tau}_{t+1}, \mathbf{a}'_{t+1}), \quad (1)$$

where $\boldsymbol{\tau}_t = (\tau_t^1, \ldots, \tau_t^N)$ is the joint history, $Q^*_{\text{targ}}$ the target, and $a_t^{i\prime} = \arg\max Q^i(\tau_t^i, \cdot)$ the individual target action. WQMIX updates $Q^{\text{tot}}$ toward the target $y_t$ by adaptively weighting the TD error with $w(s_t, \mathbf{a}_t)$, where $w(s_t, \mathbf{a}_t) = 1$ if $Q^{\text{tot}}(s_t, \boldsymbol{\tau}_t, \mathbf{a}_t) < y_t$ and $w(s_t, \mathbf{a}_t) = w_c < 1$ otherwise, thereby prioritizing updates on underestimated actions.

### 2.3 COMMUNICATION IN DEC-POMDPS

In Dec-POMDPs, partial observability frequently necessitates inter-agent communication to enable effective coordination (Sukhbaatar et al., 2016). A common approach is for each agent to transmit a message $m \in M$, where $M$ denotes the set of all possible messages and $m_t^i$ is the message received by agent $i$ at time $t$. These messages are typically incorporated into each agent's individual utility function $Q^i(\tau_t^i, a_t^i, m_t^i)$, allowing agents to leverage shared information to improve decision-making. For example, MASIA (Guan et al., 2022) employs an encoder–decoder architecture that extracts a latent representation $z_t$ from the joint observation and reconstructs the global state $s_t$ so that agents can communicate message given by functions of $z_t$. Most communication-based MARL approaches therefore require message exchange during both training and evaluation. In contrast, we consider a more flexible setting where communication is used only when necessary: evaluation can proceed without message passing in environments that do not require it, while in communication-critical tasks, agents can still exchange latent representations to coordinate effectively.

## 3 RELATED WORKS

### 3.1 CTDE METHODS IN MARL

Within the CTDE paradigm, value decomposition methods decompose the joint action-value into individual utilities, ensuring scalable MARL (Sunehag et al., 2017; Rashid et al., 2020a; Yang et al., 2020). Building on this foundation, recent works suggest more expressive identity representation (Naderializadeh et al., 2020; Zang et al., 2023; Liu et al., 2023a), or assigning roles based on agent trajectories (Wang et al., 2020b;c; Zeng et al., 2023). Another line of work extends exploration strategies studied in the single-agent setting (Tang et al., 2017; Burda et al., 2018; Han & Sung, 2021a;b) to the multi-agent domain under the CTDE paradigm, introducing centralized latent variables or coordination signals to capture richer joint behaviors (Mahajan et al., 2019; Li et al., 2021; Jo et al., 2024). In parallel, several works focus on making real-world deployment more feasible via improving the sample efficiency, (Yang et al., 2024; Qin et al., 2024) and robustness of MARL algorithms (Yuan et al., 2023; Lee et al., 2025). Finally, another line of research extends the actor-critic framework to the multi-agent setting by employing centralized critics with decentralized actors (Lowe et al., 2017; Foerster et al., 2018; Iqbal & Sha, 2019; Su et al., 2021).

### 3.2 OVERCOMING THE MONOTONICITY CONSTRAINT

While ensuring tractability, QMIX's monotonicity constraint significantly limits representational capacity, motivating active research to overcome this restriction. WQMIX (Rashid et al., 2020b) mitigates these limitations through weighted projections and non-monotonic targets. In addition, some methods leverage a centralized critic with no inherent constraints (Wang et al., 2020d; Zhang et al., 2021; Peng et al., 2021), while recent works pursue more flexible factorizations of the joint action-value function (Son et al., 2019; Wang et al., 2020a; Wan et al., 2021; Li et al., 2023b; 2024), or incorporating alternative learning objectives (Zhou et al., 2022; Hu et al., 2023; Zhou et al., 2023; Li et al., 2023a; Liu et al., 2023b). A representative example of such an objective is risk sensitivity, accounting for return variance (Qiu et al., 2021; Shen et al., 2023; Chen et al., 2024).

### 3.3 COMMUNICATION IN MARL

Communication has emerged as a critical mechanism in MARL for enabling coordination under partial observability. A range of works focus on the design of communication protocols, specifying how and when agents should exchange messages (Foerster et al., 2016; Sukhbaatar et al., 2016; Liu et al., 2020; Hu et al., 2024). Others investigate the type of information that should be shared to ensure that communication is informative (Li & Zhang, 2023; Shao et al., 2023). Another group aims to improve communication efficiency by reducing redundancy or compressing messages (Wang et al., 2019; Guan et al., 2022). Finally, selective communication strategies aim to determine which messages to send or whom to communicate with, thereby reducing unnecessary communication overhead (Das et al., 2019; Yuan et al., 2022; Zhu et al., 2024; Sun et al., 2024).

## 4 METHODOLOGY

### 4.1 MOTIVATION: OVERCOMING DYNAMIC OPTIMALITY SHIFTS IN MARL

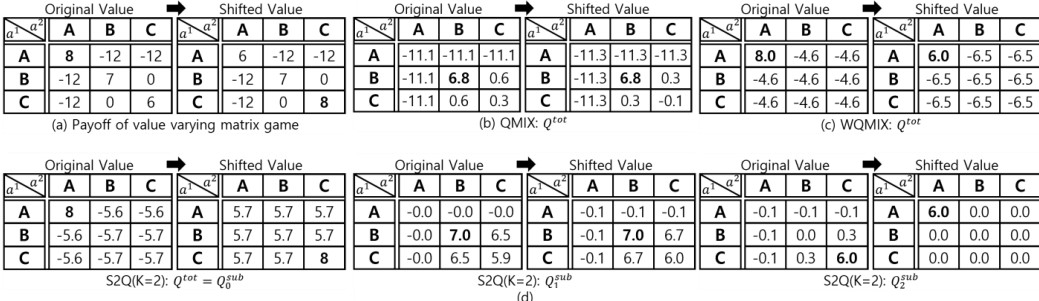

Figure 1: Fundamental Limitations of value decomposition algorithms. (a): The actual payoff of the matrix game. (b),(c): Training result of QMIX and WQMIX. (d): Training results of S2Q when $K = 2$, where $K$ is the hyperparameter controlling number of sub-networks to use.

In this section, we revisit a key limitation of value decomposition under the CTDE paradigm. QMIX is known to struggle in representing the true global value due to the IGM constraint, and several methods have been proposed to mitigate this issue (Rashid et al., 2020b). Most focus on learning an unconstrained value function (e.g., WQMIX) or enhancing state-awareness for better inference of the global optimum. Yet, a core challenge remains: by focusing solely on a single optimal joint action, these methods still fail to recover the global optimum when the value function changes dynamically.

To illustrate this issue, we consider the payoff matrix game in Fig. 1, consisting of a single state with two agents, each selecting actions from $A, B, C$. The payoff matrix specifies the joint value for each $(a^1, a^2)$. In Fig. 1(a, left), the optimal joint action is $(A, A)$ with value 8, while $(B, B)$ and $(C, C)$ are suboptimal with values 7 and 6. After training the algorithms to convergence, we modify the payoff as shown in Fig. 1(a, right), where the optimum shifts to $(C, C)$ with value 8, and $(A, A)$ and $(B, B)$ drop to 6 and 7. This setup mirrors realistic MARL scenarios in which exploration updates value estimates and causes the optimal action to change. Figures 1(b)–(c) show the learning behavior of QMIX (Rashid et al., 2020a) and WQMIX (Rashid et al., 2020b) before and after the payoff change. We use WQMIX as a representative example since the methods relying on enhanced state-awareness are expected to behave similarly to WQMIX, as the matrix game involves only a single state. The results reveal that QMIX fails in both cases because its monotonic mixing network cannot capture the non-monotonic value structure, while WQMIX uses an unconstrained target $Q^*$ and a weighted TD objective but still fails to adapt when the optimum shifts to $(C, C)$. These findings expose a key weakness, as existing methods often converge to suboptimal solutions because they are unable to track a moving optimum.

We attribute this failure to the fact that conventional methods do not explicitly track suboptimal actions. Once information about alternative high-value modes is discarded, the learner cannot adapt when those actions later become optimal. To overcome this limitation, we propose Successive Sub-value Q-learning (S2Q), a novel MARL framework that successively learns sub-value functions $Q_k^{\text{sub}}$, $k = 1, \ldots, K$, which share the same architecture as $Q^{\text{tot}}$ but are each dedicated to capturing a distinct suboptimal action. When the optimal action changes, S2Q can immediately leverage the corresponding sub-value function and guide $Q^{\text{tot}}$ to adapt. As shown in Fig. 1(d) with $K = 2$, under the original payoff matrix $Q_0^{\text{sub}} := Q^{\text{tot}}$ learns the optimal $(A, A)$, while $Q_1^{\text{sub}}$ and $Q_2^{\text{sub}}$ capture $(B, B)$ (second-optimal) and $(C, C)$ (third-optimal). After the payoff is modified, $Q_2^{\text{sub}}$ identifies $(C, C)$ as optimal, enabling $Q^{\text{tot}}$ to rapidly pivot to the correct solution. Beyond adaptability, the maintained sub-value functions also support more effective exploration than standard $\epsilon$-greedy by actively sampling alternative high-value modes. To further illustrate S2Q's exploration behavior, Appendix D presents heatmaps of joint action probabilities showing that S2Q identifies and prioritizes valuable suboptimal actions and consequently adapts quickly when the value landscape shifts. Through this design, S2Q is expected to achieve faster convergence than existing CTDE methods, demonstrating its efficiency across diverse MARL environments. The next section details the successive learning scheme of the proposed S2Q.

## 4.2 Successive Sub-value Q-learning for Retaining Suboptimal Actions

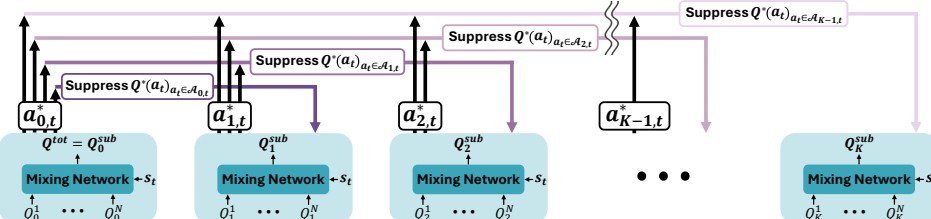

Figure 2: Illustration of S2Q framework. Each subnetwork $Q^{sub,k}$ transmits $\mathcal{A}^k$, a set of optimal actions according to all previous subnetworks. $Q^{sub,k+1}$ learns the unrestricted target $Q^*$ while suppressing the Q-values of actions included in $\mathcal{A}^k$.

We now present the detailed formulation of S2Q, introduced in Section 4.1. S2Q trains multiple sub-value functions to follow $Q^*$ as in WQMIX, but with one key idea: the first sub-value function identifies the maximum joint action under the IGM condition, and the next sub-value function suppresses its value in the learning objective. This makes the action no longer the maximum, allowing the next sub-value function to learn the highest remaining joint action under IGM, effectively the

next suboptimal action of $Q^*$. Repeating this process yields a sequence of high-value candidates, with each sub-value function specializing in a different region of the action space.

Specifically, S2Q employs a sequence of sub-value functions $Q_k^{\text{sub}}$, $k = 0, \ldots, K$ to successively track optimal and suboptimal actions. By definition, we set $Q_0^{\text{sub}} := Q^{\text{tot}}$, where $Q^{\text{tot}}$ is trained to select the true optimal action, while each $Q_k^{\text{sub}}$ aims to capture the $k$-th suboptimal action by suppressing those identified by previous sub-value functions. All sub-value functions share the same mixture architecture as QMIX and satisfy the IGM condition. Specifically, each $Q_k^{\text{sub}}$ consists of independent individual utilities $Q_k^i$, and the joint greedy action $\mathbf{a}_{k,t}^* = (a_{k,t}^{1*}, \ldots, a_{k,t}^{N*})$ with $a_{k,t}^{i*} = \arg\max Q_k^i(\tau_t^i, \cdot)$ corresponds to the $k$-th suboptimal joint action.

To optimize each $Q_k^{\text{sub}}$, we build upon the TD-learning objective of WQMIX by leveraging the unrestricted $Q^*$ trained via equation 1 and incorporating a suppression term for previously identified suboptimal actions. The resulting objective is defined as:

$$\mathbb{E}_{(s_t, \boldsymbol{\tau}_t, \mathbf{a}_t) \sim \mathcal{B}} \left[ w_k(s_t, \mathbf{a}_t) \left( Q_k^{\text{sub}}(s_t, \boldsymbol{\tau}_t, \mathbf{a}_t) - \left( y_t - \alpha \mathbb{I}(\mathbf{a}_t \in \mathcal{A}_{k-1,t}) \cdot \max(Q_{\text{targ}}^*(s_t, \boldsymbol{\tau}_t, \mathbf{a}_t), C) \right) \right)^2 \right], \tag{2}$$

where $w_k$ is a WQMIX-based weighting function for each $k$, detailed in Appendix C, $y_t = r_t + \gamma Q_{\text{targ}}^*(s_{t+1}, \boldsymbol{\tau}_{t+1}, \mathbf{a}_{0,t+1}^*)$ is the target from $Q^*$ in equation 1, $\mathbb{I}$ is the indicator function, $\mathcal{B}$ is the replay buffer, and $\mathcal{A}_{k,t} = \{\mathbf{a}_{0,t}^*, \mathbf{a}_{1,t}^*, \ldots, \mathbf{a}_{k,t}^*\}$ with $\mathcal{A}_{-1,t} = \emptyset$ denotes the set of previously identified suboptimal actions. The factor $\alpha$ controls how strongly the values of actions in $\mathcal{A}_{k-1,t}$ are suppressed, and $\max(\cdot, C)$ with a positive constant $C > 0$ is applied to properly handle potentially negative values of $Q_{\text{targ}}^*$. When $Q_{\text{targ}}^*$ is positive, we can simply use $Q_{\text{targ}}^*$ instead of $\max(Q_{\text{targ}}^*, C)$ in equation 2 for implementation. From the proposed successive Q-learning, each sub-value function $Q_k^{\text{sub}}$ can select the $k$-th suboptimal joint action of a given $Q^*$ exactly, as guaranteed by the following theorem, provided that the suppression constant $\alpha$ is sufficiently large.

**Theorem 4.1.** *Let $Q^*(s_t, \boldsymbol{\tau}_t, \mathbf{a}_t)$ and $\{Q_k^{\text{sub}}\}_{k=0}^K$ be the joint action-value function and sub-value functions obtained by minimizing equation 1 and equation 2, respectively, and let $\{\mathbf{a}_{0,t}^*, \ldots, \mathbf{a}_{K,t}^*\}$ denote the $K+1$ successive suboptimal joint actions of $Q^*$ for given $s_t$ and $\boldsymbol{\tau}_t$ at timestep $t$ s.t.*

$$Q^*(s_t, \boldsymbol{\tau}_t, \mathbf{a}_{0,t}^*) \geq \cdots \geq Q^*(s_t, \boldsymbol{\tau}_t, \mathbf{a}_{K,t}^*) \geq Q^*(s_t, \boldsymbol{\tau}_t, \mathbf{a}_t), \quad \forall \mathbf{a}_t \notin \{\mathbf{a}_{0,t}^*, \ldots, \mathbf{a}_{K,t}^*\}.$$

*If the reward function $r$ is bounded and suppression factor $\alpha$ is sufficiently large, then,*

$$\mathbf{a}_{k,t}^* = \arg\max_{\mathbf{a}_t} Q_k^{\text{sub}}(s_t, \boldsymbol{\tau}_t, \mathbf{a}_t) \quad \forall k \in \{0, ..., K\}. \tag{3}$$

**Proof)** Proof of Theorem 4.1 is provided in Appendix B.

According to Theorem 4.1, it is guaranteed that, with bounded rewards and a sufficiently large suppression factor $\alpha$, the successive learning procedure maintains $Q_0^{\text{sub}}, \ldots, Q_K^{\text{sub}}$, where $Q_0^{\text{sub}}$ represents the global optimum and the remaining sub-value functions capture successive suboptimal actions. By explicitly tracking the high-value joint actions, S2Q allows $Q^*$ to effectively update its values and ensures that $Q_0^{\text{sub}}$ can promptly adapt when the optimal action changes. Fig. 2 illustrates this process, where each $Q_k^{\text{sub}}$ suppresses previously identified actions before selecting the next best candidate. Importantly, although this procedure is presented in the context of WQMIX, it is general and can be applied to any CTDE method by replacing $Q^*$ with $Q^{\text{tot}}$. Consequently, S2Q can follow changes in the value landscape more closely and adapt to new optima faster than conventional approaches.

### 4.3 COORDINATED EXECUTION VIA COMMUNICATION DURING TRAINING

As described above, S2Q continuously tracks suboptimal actions, but for these actions to contribute effectively, $Q^*$ must be trained toward global convergence. However, prior MARL methods typically rely on agent-wise $\epsilon$-greedy exploration, under which the probability of joint exploration decreases exponentially with the number of agents, causing most joint actions to remain near the current optimum. To overcome this limitation, we explicitly execute tracked suboptimal actions with priority determined by a Softmax distribution $\mathbf{P}_t$ over their $Q^*$ values, thereby ensuring more frequent visits, enabling $Q^*$ to find better optimal and suboptimal actions, defined as

$$\mathbf{P}_t = (P_{0,t}, \ldots, P_{K,t}) := \text{Softmax}\left( Q^*(s_t, \boldsymbol{\tau}_t, \mathbf{a}_{0,t}^*)/T, \ldots, Q^*(s_t, \boldsymbol{\tau}_t, \mathbf{a}_{K,t}^*)/T \right), \tag{4}$$

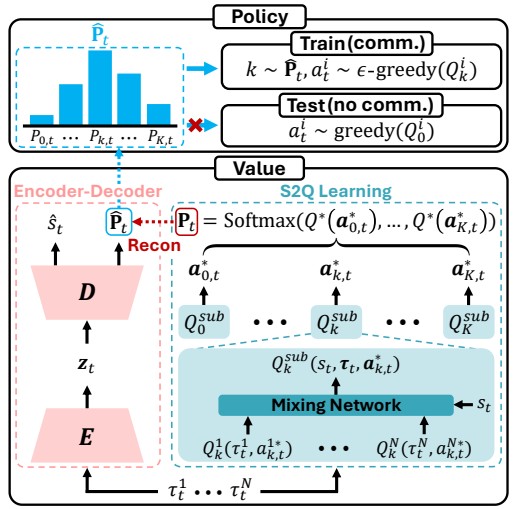

Figure 3: Overall framework of S2Q

**Algorithm 1** S2Q Framework

1: Initialize $Q^*$, $[Q_k^{sub}]_{k=0}^K$, $(E, D)$
2: **for** each training iteration **do**
3:     **for** each environment step $t$ **do**
4:         **if** training **then**
5:             Obtain $z_t = E(\boldsymbol{\tau}_t)$
6:             Obtain $\hat{\mathbf{P}}_t, \hat{s}_t = D(z_t)$
7:             Sample $k \sim \hat{\mathbf{P}}_t$
8:             Sample $\mathbf{a}_t \sim \epsilon\text{-greedy}(Q_k^{sub})$
9:         **else**
10:            Sample $\mathbf{a}_t \sim \text{greedy}(Q^{sub}, 0)$
11:        **end if**
12:    **end for**
13:    Compute target $y$
14:    Update $Q^*$, $Q_k^{sub}$ via Equation 1, 2
15:    Update $E$, $D$
16: **end for**

where $T$ is a temperature parameter. The behavior policy first samples $k \sim \mathbf{P}_t$, and then executes actions from $Q_k^{\text{sub}}$ according to $\epsilon$-greedy rule, i.e., $a_t^i \sim \epsilon\text{-greedy}(Q_k^i)$. This design first selects $k$ based on $Q^*$ and then explores around the corresponding suboptimal action, and we empirically show in Appendix D that S2Q visits a wider range of spaces than conventional $\epsilon$-greedy exploration.

Exact computation of $\mathbf{P}_t$, however, requires access to global information that decentralized agents cannot directly observe. More importantly, since all agents must select the same sub-value function index $k$ in order to execute $Q_k^{\text{sub}}$ consistently, coordination among agents is necessary. To resolve these challenges, we introduce communication during training, allowing agents to estimate $\mathbf{P}_t$ jointly and synchronize their choice of $Q_k^{\text{sub}}$. For this purpose, we adopt an encoder–decoder architecture for general representation learning, following prior communication methods (Guan et al., 2022; Wang et al., 2019). Specifically, encoder $E$ maps local histories $\boldsymbol{\tau}_t$ into a latent representation $z_t = E(\boldsymbol{\tau}_t)$, and the decoder $D$ reconstructs both the global state and an approximate distribution $(\hat{s}_t, \hat{\mathbf{P}}_t) = D(z_t)$. Agents then synchronize on the same $k$ sampled from $\hat{\mathbf{P}}_t$ instead of $\mathbf{P}_t$ and execute the corresponding $Q_k^{\text{sub}}$, ensuring consistent and accurate use of sub-value functions.

At test time, by contrast, communication is not required in the default setup of S2Q. Since $Q_0^{\text{sub}} = Q^{\text{tot}}$ alone suffices to produce the greedy optimal action $\mathbf{a}_{0,t}^*$, evaluation remains fully decentralized and communication-free. This provides a practical advantage over conventional communication methods that rely on message passing even during evaluation. Nevertheless, in environments where communication is indispensable for task success, we also consider a variant, denoted S2Q-Comm, in which the latent $z_t$ is provided to each $Q_k^i$ during both training and execution. Through this practical design, S2Q integrates successive sub-value functions with a Softmax-based behavior policy, enabling $Q^{\text{tot}}$ to adapt quickly when the optimal action changes. The overall S2Q framework is illustrated in Fig. 3, where each local history $\tau_t^i$ is processed along the S2Q Learning module, and the Encoder-Decoder module to generate the softmax probabilities $\mathbf{P}_t$, and its prediction $\hat{\mathbf{P}}_t$. During training, agents use $\hat{\mathbf{P}}_t$ to synchronize their selection of the sub-value function $Q_k^{\text{sub}}$, while evaluation relies solely on the local outputs. The training procedure is summarized in Algorithm 1, and further details on loss functions and implementation are provided in Appendix C.

## 5 EXPERIMENTS

In this section, we evaluate S2Q on two widely used MARL benchmarks: the StarCraft Multi-Agent Challenge (SMAC) (Samvelyan et al., 2019), which involves micromanagement tasks in StarCraft II, and Google Research Football (GRF) (Kurach et al., 2020), which features cooperative soccer-like scenarios with an opponent team including a goalkeeper. As shown in Fig. 4(a), we consider the SMAC-Hard+ suite, consisting of one hard map (5m_vs_6m) and five super hard maps (MMM2, 27m_vs_30m, corridor, 6h_vs_8z, 3s5z_vs_3s6z), all of which require a high degree

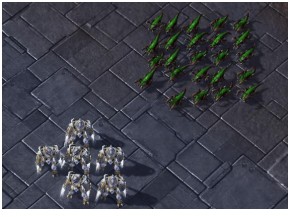
(a) SMAC-Hard+: Corridor

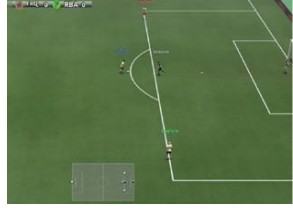
(b) GRF: academy_3_vs_2

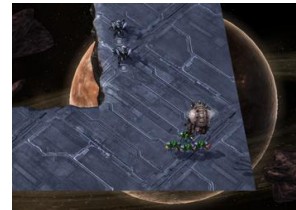
(c) SMAC-Comm: 1o_2r_vs_4r

Figure 4: Experiment environments

of coordination among agents. Fig. 4(b) illustrates the GRF setups, including `academy_3_vs_2` and `academy_4_vs_3`, where the number of agents and team formations are varied. In addition, Fig. 4(c) depicts the SMAC-Comm suite, which explicitly requires agent communication for successful task completion, including `1o_10b_vs_1r`, `1o_2r_vs_4r`, `5z_vs_1ul`, and `bane_vs_hM`. Further details of the environmental setup are provided in Appendix E. For all reported result tables and plots, we present the mean and standard deviation across five random seeds.

## 5.1 PERFORMANCE ANALYSIS

For performance evaluation, we consider communication-free tasks from `SMAC-Hard+` and `GRF`, along with communication-demanding scenarios from `SMAC-Comm`, with appropriate baselines for each setting to ensure fair comparison. In the `SMAC-Hard+` and `GRF` benchmarks, we compare S2Q with QMIX and methods addressing the limitations of monotonic value decomposition, including WQMIX[1], which leverages $Q^*$ for more accurate value estimation; DOP (Wang et al., 2020d) and FOP (Zhang et al., 2021), which extend the actor–critic paradigm to promote better global coordination; PAC (Zhou et al., 2022) and RiskQ (Shen et al., 2023), which integrate counterfactual prediction or risk-aware objectives; and MARR (Yang et al., 2024), which improves sample efficiency through reset mechanisms. We also include MASIA (Guan et al., 2022), which reconstructs global state information from local observations, to evaluate the effect of communication. For the `SMAC-Comm` scenarios, where information exchange among agents is critical under partial observability, we adopt S2Q-Comm, a variant that provides the latent $z_t$ to each agent's utility $Q^i(\tau_t^i, z_t, a_t^i)$, and compare it with QMIX and QMIX-Comm, a variant of QMIX augmented with learned latent information produced by an encoder–decoder structure that reconstructs the state, FullCom, which augments QMIX with full observation sharing, and recent communication-focused MARL methods such as MASIA, NDQ (Wang et al., 2019), and CAMA (Shao et al., 2023), which enhance communication via information-theoretic regularization or complementary attention, as well as MAIC (Yuan et al., 2022) and T2MAC (Sun et al., 2024), which employ selective and targeted communication for improved coordination efficiency. All algorithms are implemented using the authors' official code, and S2Q is trained with the best-performing hyperparameter ($K = 2$, $T = 0.1$). Since the $Q$-values quickly turn positive in environments we consider, we adopt $Q^*_{\text{targ}}$ instead of $\max(Q^*_{\text{targ}}, C)$ in equation 2 for implementation, as mentioned in Section 4.2. Additional experimental details, including the hyperparameter setup of S2Q are provided in Appendix F.

**SMAC-Hard+ and GRF**: Fig. 5 compares performance on SMAC-Hard+ and GRF. S2Q consistently outperforms existing baselines, achieving faster convergence and higher asymptotic returns. This advantage is most evident in exploration-intensive scenarios such as `6h_vs_8z` and `3s5z_vs_3s6z`, where conventional methods adapt slowly to the optimal joint action shifts, as shown in the payoff matrix experiment in Section 4.1. By continually tracking suboptimal actions with successive sub-value functions, S2Q allows $Q^{\text{tot}}$ to rapidly adjust toward the new optimum, enabling more efficient exploration and quicker convergence, even in the challenging GRF tasks.

**SMAC-Comm**: Fig. 6 presents results for communication-critical tasks. S2Q-Comm performs comparably to baselines in the simpler `1o_2r_vs_4r` scenario, but demonstrates substantial gains in more challenging settings such as `5z_vs_1ul` and `bane_vs_hM`, where tight coordination and real-time adaptation are required. Synchronized sub-value function selection through the learned

---

[1]Throughout our experiments, we adopt OW-QMIX variant of WQMIX (Rashid et al., 2020b).

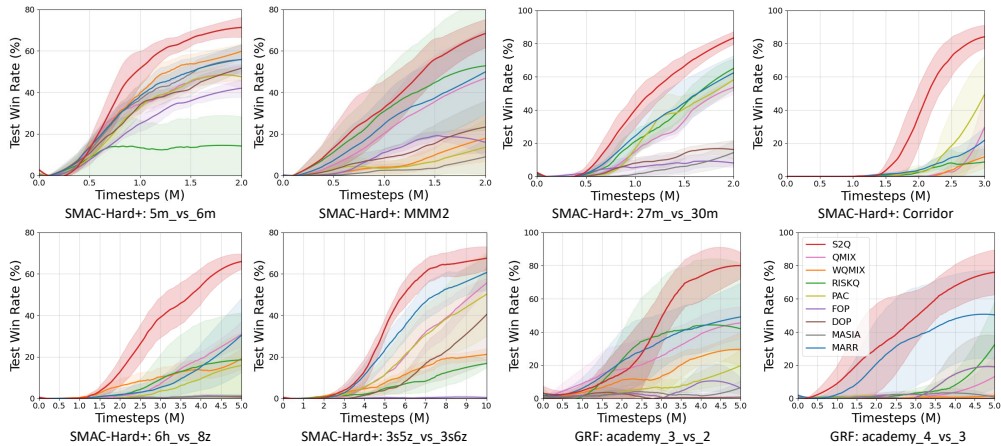

Figure 5: Performance comparison: Average test win rates in the SMAC-Hard+ and GRF tasks

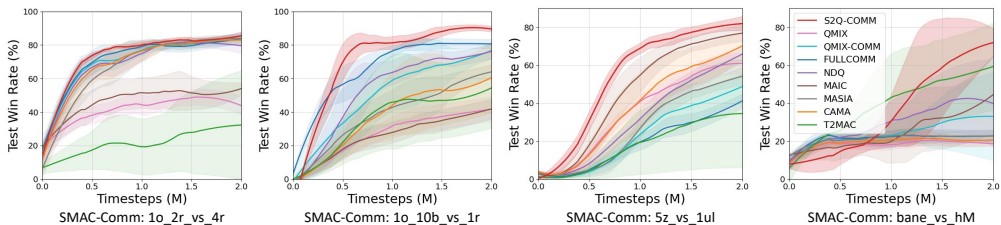

Figure 6: Performance comparison: Average test win rates in the SMAC-Comm tasks

latent $z_t$ enables agents to consistently exploit informative suboptimal actions, leading to more efficient cooperation. In addition, comparison with QMIX-Comm shows that these improvements arise from S2Q's successive sub-value learning, rather than relying solely on information sharing. These findings suggest that S2Q-Comm can serve as a general approach for leveraging communication efficiently, providing a promising direction for scaling to larger teams and more complex partially observable environments.

Together, these experiments show that by continually tracking and exploiting suboptimal actions, S2Q achieves faster convergence and more efficient learning than state-of-the-art MARL methods. We further evaluate S2Q on additional stochastic environments, including SMACv2 (Ellis et al., 2023), which randomizes team compositions, and SMAC-Hard (Deng et al., 2024), with mixed scripted opponents. For clarity, we refer to the latter benchmark as SMAC-Hard (Mixture Opponent) to distinguish it from the `SMAC-Hard+` environment. Despite the increased stochasticity, S2Q consistently outperforms all baselines, suggesting that our method can effectively address stochastic dynamics to a meaningful extent. Furthermore, we extend S2Q to other CTDE methods such as VDN (Sunehag et al., 2017) and QPLEX (Wang et al., 2020a), observing similarly consistent performance gains. Detailed results for these additional experiments are provided in Appendix G.

## 5.2 TRAJECTORY ANALYSIS

To investigate the training dynamics of S2Q ($K = 2$), we analyze the distribution of agents' actions in the `6h_vs_8z` scenario, shown in Fig. 7(a). Fig. 7(b) reports the evolution of hit rate and win rate over training. Since SMAC tasks require agents to defeat the opposing team, the dominant actions are *move* and *hit*. Early in training, agents tend to prioritize survival, favoring *move* and exhibiting low hit rates. However, as training progresses, the hit rate gradually increases, leading to higher win rates, indicating that the optimal policy shifts from evasive behavior towards more aggressive strategies where *hit* dominates. Fig. 7(c) illustrates how S2Q adapts to this shift in optimality by leveraging its successive sub-value functions. Initially, *move* is considered the optimal action ($\mathbf{a}_{0,t}^*$), while *hit* actions ($\mathbf{a}_{1,t}^*, \mathbf{a}_{2,t}^*$) are tracked as suboptimal. As training proceeds and $Q^*$

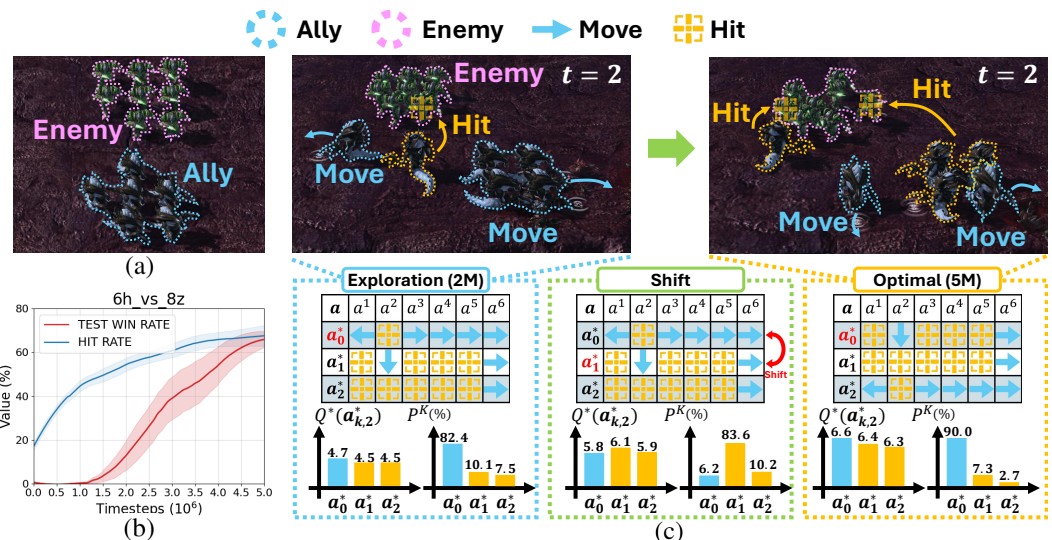

Figure 7: Training behavior of S2Q. Left: training step = 2M. Right: training step = 5M

recognizes that *hit* yields higher returns, S2Q gradually increases the execution frequency of *hit* via the Softmax-based behavior policy. This enables $Q_0^{\text{sub}}$ to promptly adjust to the newly optimal action, resulting in an increase in hit rate and ultimately higher win rates. These results provide a clear illustration of how S2Q tracks suboptimal actions and uses them to adapt efficiently when the optimal action changes. Additional trajectory analysis for the `SMAC-Comm`, `SMAC-Hard (Mixture Opponent)`, and `GRF` environments are also provided in Appendix H.4. While detailed analyses are provided in Appendix H.4, we observe a consistent trend across environments: agents frequently begin by selecting locally greedy actions conditioned on their initial states, but gradually transition toward the true optimal actions as training progresses.

## 5.3 ABLATION STUDY

In this section, we present ablation studies evaluating the contributions of S2Q's core components and hyperparameters. Component analysis is averaged across all scenarios in SMAC-Hard+, while hyperparameter analysis is conducted on `6h_vs_8z`, where S2Q's benefits are most evident. Additional results, including additional hyperparameter analysis in `MMM2` and `academy_4_vs_3`, which exhibit trends consistent with those observed in `6h_vs_8z`, ablations on the suppression constant $\alpha$, the weighting factor $w_c$ for WQMIX-based TD learning, and computational complexity, are reported in Appendix H, to validate the effectiveness and generality of S2Q across diverse settings.

Table 1: Component evaluation of S2Q on SMAC-Hard+ tasks.

| Method | Avg. Win Rate(%) |
|---|---|
| S2Q | $73.43 \pm 5.29$ |
| S2Q_oracle | $\mathbf{77.47 \pm 4.32}$ |
| S2Q_independent | $46.22 \pm 8.20$ |
| S2Q_no_wTD | $70.59 \pm 4.78$ |
| S2Q_no_soft | $55.17 \pm 6.71$ |
| S2Q_random | $48.05 \pm 9.37$ |
| QMIX | $43.94 \pm 10.06$ |

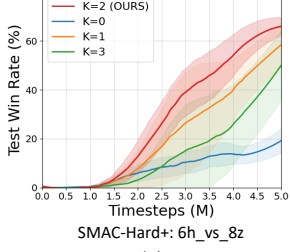

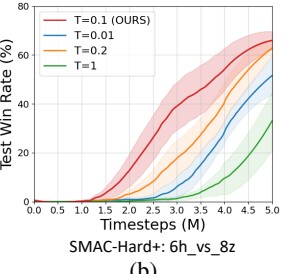

Figure 8: Hyperparameter analysis. Effect of (a) the number of sub-networks $K$ and (b) Softmax temperature $T$.

**Component evaluation**: To evaluate the contribution of each component in S2Q, we design five ablations: **S2Q_oracle** uses the true softmax distribution $\mathbf{P}_t$ instead of the estimated $\hat{\mathbf{P}}_t$; **S2Q_independent** samples $k$ independently, removing coordinated execution; **S2Q_no_wTD** re-

moves the weighted TD learning by replacing $Q^*$ with $Q_0^{\text{sub}}$; **S2Q_no_soft** eliminates Softmax-based execution, always acting according to $Q_0^{\text{sub}}$; and **S2Q_random** uniformly samples $k$ rather than using Softmax probabilities. Table 1 reports the average win rate across six SMAC-Hard+ scenarios.

The results show that 'S2Q_oracle' serves as a reference upper bound, since it can access to the true softmax distribution **P** and therefore achieves higher performance than S2Q. In contrast, 'S2Q_independent' exhibits a substantial performance drop relative to S2Q because it cannot accurately estimate the selection probabilities and consequently fails to maintain proper synchronization across agents. Together, these ablations clearly demonstrate the critical role of accurately estimating the softmax probabilities in enabling effective coordinated sub-value selection. In addition, 'S2Q_no_wTD' maintains strong performance, indicating that successive sub-value learning and its execution are more critical to performance than the auxiliary weighting scheme. In contrast, 'S2Q_no_soft' suffers a noticeable drop in win rate, showing that simply retaining suboptimal actions without prioritizing their execution is insufficient. 'S2Q_random' further degrades performance, as ignoring the relative importance of different sub-value functions dilutes learning. These findings highlight that the proposed successive sub-value learning and the Softmax-guided execution are indispensable for enabling S2Q to achieve rapid convergence toward high-quality joint policies.

**Number of sub-value functions** $K$: Controls the number of sub-value functions to learn. Fig. 8(a) shows performance for $K \in [0, 1, 2, 3]$. The results demonstrate that even with relatively small values of $K$, S2Q can effectively address diverse suboptimal actions in complex environments with large joint action spaces, such as SMAC, yielding rapid convergence and high final win rates. In particular, $K = 2$ achieves the best performance, reaching nearly 70% win rate. On the other hand, setting $K = 0$ fails to capture meaningful suboptimal actions, thereby limiting the effectiveness of exploration. Conversely, a larger value, such as $K = 3$, may introduce excessive variance and destabilize learning. These findings indicate that a moderate number of candidate sub-networks achieves a favorable trade-off between diversity and stability, maximizing the effectiveness of S2Q.

**Softmax temperature** $T$: The temperature $T$ regulates the sharpness of the Softmax distribution $\mathbf{P}_t$. Fig. 8(b) shows the performance for $T \in \{0.01, 0.1, 0.2, 1.0\}$. The results indicate that $T = 0.1$ yields the best balance between convergence speed and final win rate. A very small temperature ($T = 0.01$) produces overly deterministic behavior early in training, limiting exploration and slowing progress toward the optimum. In contrast, a high temperature ($T = 1.0$) promotes excessive exploration, delaying convergence and occasionally leading to suboptimal plateaus. Interestingly, $T = 0.2$ also performs competitively, suggesting that S2Q is relatively robust to moderate changes in temperature, as long as sampling remains focused enough to prioritize promising sub-values.

## 6    LIMITATIONS

While S2Q shows strong performance across diverse MARL environments, using multiple sub-value functions increases computation and memory requirements. However, our analysis in Appendix H.5 shows that this overhead is moderate relative to the performance gains, making the trade-off favorable in most practical settings where faster convergence and higher final performance are critical. In addition, S2Q relies on a Softmax-based selection strategy with a temperature parameter controlling the exploration-exploitation balance. Nevertheless, we find that S2Q is not highly sensitive to this parameter, and a reasonable value can be chosen with minimal tuning effort, allowing S2Q to remain practical and easily deployable across a wide range of environments.

## 7    CONCLUSION

In this work, we addressed the fundamental challenge posed by monotonic value decomposition in MARL and introduced S2Q, a novel framework based on successive Q-learning. By sequentially learning multiple sub-value functions, S2Q continuously tracks high-value alternative actions and rapidly adapts when the optimal joint action shifts, leading to more effective exploration and faster convergence. This design enables efficient coordination without requiring explicit communication or access to global state information. Extensive experiments on challenging MARL benchmarks, including SMAC and GRF, demonstrate that S2Q consistently avoids suboptimal solutions induced by monotonicity constraints and achieves substantial improvements over state-of-the-art methods. These results position S2Q as a practical and general framework for advancing cooperative MARL.

ACKNOWLEDGMENT

This work was supported partly by the Institute of Information & Communications Technology Planning & Evaluation (IITP) grant funded by the Korea government (MSIT) (No. RS-2022-II220469, Development of Core Technologies for Task-oriented Reinforcement Learning for Commercialization of Autonomous Drones), (No. RS-2025-25442824, AI Star-Fellowship Program (UNIST)), and (No. RS-2020-II201336, Artificial Intelligence Graduate School Support (UNIST)), and partly by the National Research Foundation of Korea (NRF) grant funded by the Korea government (MSIT) (No. RS-2025-23523191, LLM-Based Multi-Agent Reinforcement Learning for End-to-End Large Autonomous Swarm Control).

ETHICS STATEMENT

This work focuses on algorithmic improvements in cooperative multi-agent reinforcement learning (MARL) under the CTDE paradigm. Our study uses publicly available simulation environments, including SMAC (Samvelyan et al., 2019), Google Research Football (Kurach et al., 2020), and SMACv2 (Ellis et al., 2023), which do not involve human subjects, personal data, or sensitive information. Our work is motivated by cooperative and coordination tasks, and does not explicitly encourage or evaluate harmful uses. We adhere to the ICLR Code of Ethics and affirm that all research was conducted in alignment with principles of fairness, transparency, and academic integrity.

REPRODUCIBILITY STATEMENT

To ensure reproducibility, we provide a detailed description of S2Q in Section 4.3, with implementation details in Appendix C. Benchmarks including SMAC (Samvelyan et al., 2019), GRF (Kurach et al., 2020), and SMACv2 (Ellis et al., 2023) are publicly available, with descriptions and code links in Appendix E. An anonymized code repository containing the full implementation is submitted as supplementary material where configurations are detailed in Appendix F. Finally, considered baselines and their repositories are listed in Appendix F.3. Together, these resources collectively enable independent reproduction of our results reported in this paper.

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

## A  THE USE OF LARGE LANGUAGE MODELS (LLMs)

In preparing this paper, we made limited use of large language models (LLMs) exclusively for polishing grammar and improving the clarity of our writing. No part of the research process, including problem formulation, method design, experimentation, analysis, or interpretation, was conducted using LLMs. Their role was strictly restricted to editorial assistance, ensuring the presentation of our work is clear and readable.

## B  MISSING PROOF

**Theorem B.1** (Successive Q-learning). *Let $Q^*(s_t, \boldsymbol{\tau}_t, \mathbf{a}_t)$ and $\{Q_k^{\mathrm{sub}}\}_{k=0}^K$ be the joint action-value function and sub-value functions obtained by minimizing equation 1 and equation 2, respectively, and let $\{\mathbf{a}_{0,t}^*, \ldots, \mathbf{a}_{K,t}^*\}$ denote the $K+1$ successive suboptimal joint actions of $Q^*$ for given $s_t$ and $\boldsymbol{\tau}_t$ at timestep $t$ s.t.*

$$Q^*(s_t, \boldsymbol{\tau}_t, \mathbf{a}_{0,t}^*) \geq \cdots \geq Q^*(s_t, \boldsymbol{\tau}_t, \mathbf{a}_{K,t}^*) \geq Q^*(s_t, \boldsymbol{\tau}_t, \mathbf{a}_t), \quad \forall \mathbf{a}_t \notin \{\mathbf{a}_{0,t}^*, \ldots, \mathbf{a}_{K,t}^*\}.$$

*If the reward function $r$ is bounded and suppression factor $\alpha$ is sufficiently large, then,*

$$\mathbf{a}_{k,t}^* = \arg\max_{\mathbf{a}_t} Q_k^{\mathrm{sub}}(s_t, \boldsymbol{\tau}_t, \mathbf{a}_t) \quad \forall k \in \{0, ..., K\}. \tag{B.1}$$

*Proof.* We consider the minimization of the joint value loss equation 1 together with the proposed sub-value losses equation 2. At convergence, the Bellman optimality equation implies that $Q^*$ satisfies $y_t = r_t + \gamma Q_{\mathrm{targ}}^* = Q^*$. Using this relation, each sub-value function $Q_k^{\mathrm{sub}}$ satisfies

$$Q_k^{\mathrm{sub}}(s_t, \boldsymbol{\tau}_t, \mathbf{a}_t) = \begin{cases} Q^*(s_t, \boldsymbol{\tau}_t, \mathbf{a}_t) - \alpha \max(Q^*(s_t, \boldsymbol{\tau}_t, \mathbf{a}_t), C) & \text{if } \mathbf{a}_t \in \mathcal{A}_{k-1,t}, \\ Q^*(s_t, \boldsymbol{\tau}_t, \mathbf{a}_t) & \text{otherwise,} \end{cases}$$

for every $(s_t, \boldsymbol{\tau}_t, \mathbf{a}_t)$ pairs.

For given $(s_t, \boldsymbol{\tau}_t)$ pair, if the suppression factor $\alpha$ satisfies

$$\alpha \geq \max_{\tilde{\mathbf{a}}_t \in \mathcal{A}_{k-1,t}} \frac{Q^*(s_t, \boldsymbol{\tau}_t, \tilde{\mathbf{a}}_t) - Q^*(s_t, \boldsymbol{\tau}_t, \mathbf{a}_{k,t}^*)}{\max(Q^*(s_t, \boldsymbol{\tau}_t, \tilde{\mathbf{a}}_t), C)} \geq 0, \tag{B.2}$$

then $Q_k^{\mathrm{sub}}(s_t, \boldsymbol{\tau}_t, \tilde{\mathbf{a}}_t) = Q^*(s_t, \boldsymbol{\tau}_t, \tilde{\mathbf{a}}_t) - \alpha \max(Q^*(s_t, \boldsymbol{\tau}_t, \tilde{\mathbf{a}}_t), C) \leq Q^*(s_t, \boldsymbol{\tau}_t, \mathbf{a}_{k,t}^*) = Q_k^{\mathrm{sub}}(s_t, \boldsymbol{\tau}_t, \mathbf{a}_{k,t}^*), \quad \forall \tilde{\mathbf{a}}_t \in \mathcal{A}_{k-1,t}$.

From the definition of $a_{k,t}^*$, $Q_k^{\mathrm{sub}}(s_t, \boldsymbol{\tau}_t, \tilde{\mathbf{a}}_t) = Q_k^*(s_t, \boldsymbol{\tau}_t, \tilde{\mathbf{a}}_t) \leq Q^*(s_t, \boldsymbol{\tau}_t, \mathbf{a}_{k,t}^*) = Q_k^{\mathrm{sub}}(s_t, \boldsymbol{\tau}_t, \mathbf{a}_{k,t}^*), \forall \tilde{\mathbf{a}}_t \in \mathcal{A} \backslash \mathcal{A}_{k-1,t}$, so we can conclude

$$\mathbf{a}_{k,t}^* = \arg\max_{\mathbf{a}_t} Q_k^{\mathrm{sub}}(s_t, \boldsymbol{\tau}_t, \mathbf{a}_t). \tag{B.3}$$

Thus, if we take the supremum over all state–observation histories and the maximum over all $k \in 0, \ldots, K$, the suppression factor must satisfy

$$\alpha \geq \sup_{s_t, \boldsymbol{\tau}_t} \max_{k \in \{0, \cdots, K\}} \max_{\tilde{\mathbf{a}}_t \in \mathcal{A}_{k-1,t}} \frac{Q^*(s_t, \boldsymbol{\tau}_t, \tilde{\mathbf{a}}_t) - Q^*(s_t, \boldsymbol{\tau}_t, \mathbf{a}_{k,t}^*)}{\max(Q^*(s_t, \boldsymbol{\tau}_t, \tilde{\mathbf{a}}_t), C)} \geq 0$$

, and this supremum is finite because the bounded-reward assumption implies that $Q^*$ is also bounded, which ensures that equation B.3 holds for all $(s_t, \boldsymbol{\tau}_t)$ pairs and for all $k \in \{0, \ldots, K\}$. This completes the proof of the theorem.

□

## C  IMPLEMENTATION DETAILS

In this section, we detail the implementation of the proposed S2Q method. First, we present the encoder–decoder architecture described in Section 4.3 together with its associated loss function in Appendix C.1. Then, we introduce the parameterization for the practical implementation of S2Q and the corresponding total loss function in Appendix C.2. Furthermore, details on re-defined weighting function $w_k$, which enables S2Q to more effectively shift $Q^{\mathrm{tot}}$ towards the optimal policy, are provided in Appendix C.3.

## C.1 TRAINING THE ENCODER-DECODER STRUCTURE

As introduced in Section 4.3, exact computation of $\mathbf{P}_t$ requires access to the global state $s$. Therefore, we proposed the approximation of $\mathbf{P}_t$ based on an encoder-decoder architecture. For practical implementation, we parameterize the encoder-decoder structure with neural network parameter $\psi$ and represent it as $E_\psi$ and $D_\psi$. The encoder $E_\psi$ utilizes an rnn layer, and maps the concatenation of all agents' local histories $\tau_t^i$ into a latent representation $z_t = E_\psi(\boldsymbol{\tau_t})$. Then, the decoder $D_\psi$ reconstructs both the global state and an approximate distribution $(\hat{s}_t, \hat{\mathbf{P}}_t) = D_\psi(z_t)$. Then, the encoder-decoder architecture is trained to approximate $\mathbf{P}^K$ based on a cross-entropy (CE) loss $\text{CE}(\mathbf{P}^K, \hat{\mathbf{P}}^K)$, while the reconstruction mean square error (MSE) loss $\text{MSE}(\mathbf{s}, \hat{\mathbf{s}})$ encourages $z$ to capture essential state information, thereby improving the prediction accuracy of $\hat{\mathbf{P}}^K$. The overall loss of the encoder-decoder architecture $\mathcal{L}_{\text{latent}}(\psi)$ is defined as:

$$\mathcal{L}_{\text{latent}}(\psi) = \text{CE}(\mathbf{P}^K, \hat{\mathbf{P}}^K) + \text{MSE}(\mathbf{s}, \hat{\mathbf{s}}). \tag{C.1}$$

## C.2 PRACTICAL IMPLEMENTATION OF S2Q

For practical implementation of S2Q, we parameterize both the monotonic sub-value functions $Q_0^{\text{sub}}, \ldots, Q_K^{\text{sub}}$ and the unrestricted value function $Q^*$ using neural network parameter $\theta$. As described in Section 2.2, WQMIX (Rashid et al., 2020b) trains the unrestricted value function $Q^*$ via temporal-difference (TD) loss in equation 1, which we rewrite for the parameterisized $Q$-function as:

$$\mathcal{L}_{Q^*}(\theta) = \mathbb{E}_{(s_t, \boldsymbol{\tau}_t, \mathbf{a}_t) \sim \mathcal{B}} \left[ (Q_\theta^*(s_t, \boldsymbol{\tau}_t, \mathbf{a}_t) - y_t)^2 \right], \quad y_t = r_t + \gamma Q_{\bar{\theta}}^*(s_{t+1}, \boldsymbol{\tau}_{t+1}, \mathbf{a}_{t+1}'), \tag{C.2}$$

where $\boldsymbol{\tau}_t = (\tau_t^1, \ldots, \tau_t^N)$ is the joint history, $Q_{\bar{\theta}}^*$ the target network, and $a_t^{i\prime} = \arg\max Q^i(\tau_t^i, \cdot)$ the individual target action. Then, WQMIX further updates $Q^{\text{tot}}$ toward the target $y_t$ by adaptively weighting the TD error using $w(s_t, \mathbf{a}_t)$, defined as $w(s_t, \mathbf{a}_t) = 1$ if $Q^{\text{tot}}(s_t, \boldsymbol{\tau}_t, \mathbf{a}_t) < y_t$ and $w(s_t, \mathbf{a}_t) = w_c < 1$ otherwise, thereby prioritizing updates on actions that are underestimated.

Following this design, we train $Q_\theta^*$, and then redefine the mean squared errors of the sub-value functions in equation 2 as the successive loss $\mathcal{L}_{\text{successive}}(\theta)$, expressed as:

$$\mathcal{L}_{\text{successive}}(\theta) = \sum_{k=0}^{K} \mathbb{E}_{(s_t, \boldsymbol{\tau}_t, \mathbf{a}_t) \sim \mathcal{B}} \Bigg[ w_k(s_t, \mathbf{a}_t) \Big( Q_{\theta,k}^{\text{sub}}(s_t, \boldsymbol{\tau}_t, \mathbf{a}_t)$$
$$- \big( y_t - \alpha \, \mathbb{I}(\mathbf{a}_t \in \mathcal{A}_{k-1,t}) \cdot \max(Q_{\bar{\theta}}^*(s_t, \boldsymbol{\tau}_t, \mathbf{a}_t), C) \big) \Big)^2 \Bigg], \tag{C.3}$$

where $w_k$ is a weighting function for each $k$ inspired by WQMIX, which we detail in Appendix C.3. $\mathbb{I}$ is the indicator function, $\mathcal{B}$ is the replay buffer, and $\mathcal{A}_{k,t} = \{\mathbf{a}_{0,t}^*, \mathbf{a}_{1,t}^*, \ldots, \mathbf{a}_{k,t}^*\}$ with $\mathcal{A}_{-1,t} = \emptyset$ denotes the set of previously identified suboptimal actions. The factor $\alpha$ controls how strongly the values of actions in $\mathcal{A}_{k-1,t}$ are suppressed. Combined with the encoder-decoder loss $L_{latent}(\psi)$ introduced in Appendix C.1, the final loss function of S2Q is derived as:

$$\mathcal{L}_{\text{S2Q}}(\theta, \psi) = \mathcal{L}_{\text{successive}}(\theta) + \mathcal{L}_{Q^*}(\theta) + \mathcal{L}_{\text{latent}}(\psi) \tag{C.4}$$

To further stabilize optimal action selection, we adopt a simple sampling strategy that balances stability and exploration. Specifically, with probability $p = 0.5$, the sub-network index is fixed to $k = 0$ for the entire episode, ensuring consistent learning of the global optimum. Otherwise, $k$ is sampled at each timestep from $\hat{\mathbf{P}}_t$, to promote exploration of suboptimal actions. This simple yet effective mechanism provides both reliable convergence toward the optimum and sufficient exploration diversity. Algorithm C.1 summarizes the full S2Q learning process.

---

**Algorithm C.1** Successive Sub-value Q-learning (S2Q) Framework

---

1: Initialize unrestricted value network $Q_\theta^*$ and its target network $Q_{\bar{\theta}}^*$, sub-value networks $[Q_{\theta,k}^{\text{sub}}]_{k=0}^K$, encoder-decoder $(E_\psi, D_\psi)$, and replay buffer $\mathcal{B}$
2: **for** each training iteration **do**
3:     With probability $p = 0.5$, fix sub-value index $k = 0$ for the entire episode
4:     **for** each environment step $t$ **do**
5:         **if** training **then**
6:             Encode trajectories $\boldsymbol{\tau}_t$ into latent variable $z_t = E_\psi(\boldsymbol{\tau}_t)$
7:             Decode $z_t$ to obtain categorical distribution and reconstructed state: $\hat{\mathbf{P}}_t, \hat{s}_t = D_\psi(z_t)$
8:             **if** episode is fixed to $k = 0$ **then**
9:                 Set $k = 0$
10:            **else**
11:                Sample $k \sim \hat{\mathbf{P}}_t$
12:            **end if**
13:             Select joint action $\mathbf{a}_t \sim \epsilon\text{-greedy}(Q_k^{\text{sub}})$
14:         **else**
15:             Select joint action $\mathbf{a}_t \sim \text{greedy}(Q_0^{\text{sub}})$
16:         **end if**
17:         Execute joint action $\mathbf{a}_t$, observe reward $r_t$ and next state $s_{t+1}$
18:         Store transition $(s_t, \mathbf{a}_t, r_t, s_{t+1})$ in replay buffer $\mathcal{D}$
19:     **end for**
20:     Compute TD target $y = r + \gamma Q_{\bar{\theta}}^*(s, \boldsymbol{\tau}, \mathbf{a})$
21:     Update $[Q_{\theta,k}^{\text{sub}}]_{k=0}^K$ and $Q_\theta^*$ via $\mathcal{L}_{\text{successive}}$ and $\mathcal{L}_{Q^*}$ in equations C.3, C.2
22:     Update encoder and decoder $(E, D)$ via $\mathcal{L}_{\text{latent}}$ in equation C.1
23: **end for**
24: Periodically update target network $Q_{\bar{\theta}}^*$

---

### C.3   Weighting Function in Weighted TD Learning

As explained in Section 2.2, WQMIX updates $Q^{\text{tot}}$ toward the target $y_t$ by adaptively weighting the TD error using $w(s_t, \mathbf{a}_t)$. Building on WQMIX, S2Q introduces weighting function $w_k$, defined as:

$$w_k(s_t, \mathbf{a}_t) = \begin{cases} 1, & \text{if } Q^*(s_t, \boldsymbol{\tau}_t, \mathbf{a}_t) \geq \max_{\mathbf{a}_t^* \in \mathcal{A}_K, t} Q^*(s_t, \boldsymbol{\tau}_t, \mathbf{a}_t^*), \quad k = 0 \\ 1, & \text{if } Q_k^{\text{sub}}(s_t, \boldsymbol{\tau}_t, \mathbf{a}_t) < y_t - \alpha \mathbb{I}(\mathbf{a}_t \in \mathcal{A}_{k-1,t}) \cdot Q_{\text{targ}}^*(s_t, \boldsymbol{\tau}_t, \mathbf{a}_t) \quad k = 1, \dots, K \\ w_c, & \text{otherwise.} \end{cases}$$

$$\text{(C.5)}$$

Our design of $w_k$ is motivated by the following rationale. For $k = 0$, the weighting rule ensures that optimality is consistently propagated into $Q_0^{\text{sub}} := Q^{\text{tot}}$, allowing S2Q to rely solely on $Q_0^{\text{sub}}$ during evaluation without requiring communication. For $k \geq 1$, the rule closely resembles the WQMIX weighting scheme, but instead of directly comparing against the TD target $y_t$, it suppresses the values of previously identified suboptimal actions before applying the comparison, thereby enabling the successive extraction of alternative high-value actions. In all other cases, the factor $w_c$ acts as a down-weighting term to moderate updates. A sensitivity analysis of $w_c$ is provided in Appendix H.2.

# D    SOFTMAX-BASED EXPLORATION BEHAVIOR OF S2Q

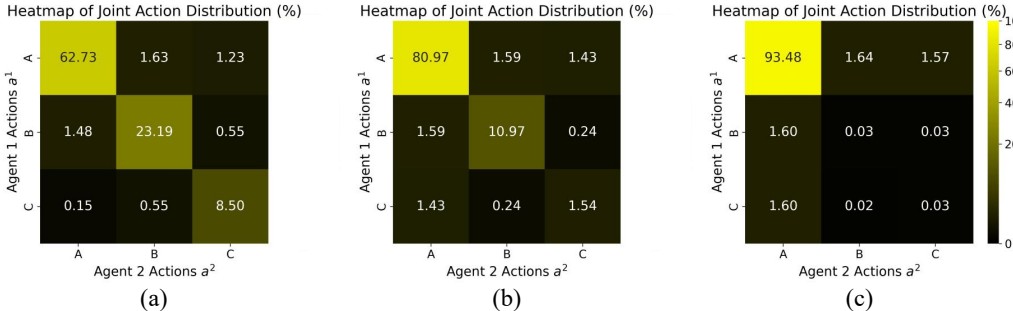

Figure D.1: Heatmap of joint action distribution under three settings: (a) S2Q ($K = 2, T = 1.0$), (b) S2Q ($K = 2, T = 0.5$), (c) per-agent $\epsilon$-greedy

In this section, we analyze the exploration behavior of S2Q compared to conventional per-agent $\epsilon$-greedy strategies. To visualize the exploration behavior of each method, we revisit the 2-agent, 3-action matrix game introduced in Section 4.1, where the optimal joint action is $(A, A)$ with value 8, and the suboptimal actions $(B, B)$ and $(C, C)$ have values 7 and 6, respectively. Fig. D.1 illustrates the joint action probabilities under three different settings: (a) S2Q with $K = 2$ and $T = 1$, (b) S2Q with $K = 2$ and $T = 0.5$, and (c) conventional per-agent $\epsilon$-greedy exploration.

Figure D.1 demonstrates that S2Q successfully identifies the optimal action while still executing suboptimal actions with meaningful frequency, even at a modest temperature of $T = 0.5$. At a higher temperature of $T = 1$, the optimal action remains the most frequently executed, while exploration across suboptimal actions is maximized. In contrast, conventional per-agent $\epsilon$-greedy exploration concentrates almost exclusively on the optimal action $(A, A)$. This highlights a key limitation of conventional methods that rely on a single optimal action and independent $\epsilon$-greedy sampling: they struggle to adapt when the value function changes dynamically. By contrast, S2Q adapts smoothly to changes in the optimal action, based on its prioritized Softmax-based exploration over tracked suboptimal actions. This broader exploration provides richer training signals for $Q^*$, facilitating faster convergence toward globally optimal solutions.

# E  ENVIRONMENT DETAILS

In this section, we provide a detailed description of the environments and scenarios used for evaluating the proposed S2Q. We consider two main benchmarks: the StarCraft Multi-Agent Challenge (SMAC) (Samvelyan et al., 2019) and the Google Research Football (GRF) environment (Kurach et al., 2020). As mentioned in Section 5, we further categorize SMAC scenarios based on their complexity and communication requirements. Specifically, we distinguish between SMAC-Hard+, which presents challenging coordination tasks with a larger number of units and complex combat strategies, and SMAC-Comm, which emphasizes tasks requiring explicit inter-agent communication for effective execution. Appendices E.1, E.2, E.3 provide detailed descriptions of the SMAC-Hard+, GRF, and SMAC-Comm scenarios discussed in Section 5. In Appendices E.4 and E.5, we detail SMACv2 (Ellis et al., 2023) and SMAC-Hard (Mixture Opponent) (Deng et al., 2024), where we conduct additional experiments to further demonstrate the effectiveness of S2Q. The performance results for SMACv2 and SMAC-Hard (Mixture Opponent) are provided in Appendices G.1 and G.2, respectively.

## E.1  THE STARCRAFT MULTI-AGENT CHALLENGE (SMAC)-HARD+

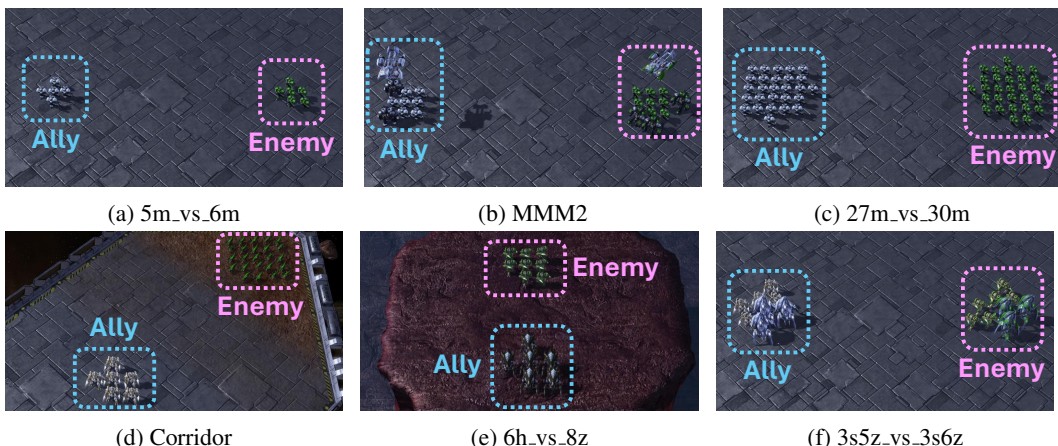

|  |  |  |
|---|---|---|
| (a) 5m_vs_6m | (b) MMM2 | (c) 27m_vs_30m |
| (d) Corridor | (e) 6h_vs_8z | (f) 3s5z_vs_3s6z |

Figure E.1: Visualizations of SMAC-Hard+ scenarios

SMAC (Samvelyan et al., 2019) is a widely used benchmark designed to test cooperative Multi-Agent Reinforcement Learning (MARL) algorithms in complex, decentralized settings. Built on top of the StarCraft II game engine, SMAC presents a series of tactical combat scenarios, where a team of AI-controlled allies faces off against enemies run by a built-in script. Each scenario differs in terms of terrain, unit types, and strategic difficulty, requiring agents to master sophisticated combat tactics such as focus fire, kiting, and exploiting environmental features. A match concludes either when one side is eliminated or the time limit expires. We provide visualizations of SMAC-Hard+ scenarios in Fig. E.1, and summarize the details of each considered episodes in Table E.1, followed by a detailed description of the SMAC benchmark.

**State space**: Global state $s$ aggregates detailed information from all entities on the battlefield. For allied units, this includes their positions, health, cooldowns, shields if applicable, and unit types. Enemy data is similar, except it omits the cooldown stats. In addition, the most recent action taken by each agent is recorded as a one-hot encoded vector.

**Observation space**: In SMAC, agents' observations are restricted to allies and enemies within a sight range of 9 units. Specifically, an agent's observation vector is composed of four distinct segments. First, movement capabilities are encoded across four directions (up, down, left, right). Second, data about visible enemies includes their relative positions, distances, health, shield values, unit type, and whether they can currently be targeted. Third, ally-related information mirrors the enemy format but excludes the agent itself. Finally, self-features reflect the observing agent's own condition:its current health, shield level (if any), and unit classification.

**Action space**: Agents in SMAC operate using a set of discrete actions. These include movement in the four primary directions (north, south, east, west), the ability to attack enemies within a range of 6 units, and special abilities limited to certain units such as Medivacs. Agents can also issue a stop command or perform a no-op action, though the latter is reserved for units that have been eliminated.

**Reward function**: SMAC incorporates a shaped reward function composed of three main elements: damage inflicted on enemy units, elimination of those units, and overall victory in the scenario. The reward is formally defined as:

$$R = \sum_{e \in \text{enemies}} \Delta\text{Health}(e) + \sum_{e \in \text{enemies}} \mathbb{I}(\text{Health}(e) = 0) \cdot \text{Reward}_{\text{death}} + \mathbb{I}(\text{win}) \cdot \text{Reward}_{\text{win}} \quad \text{(E.1)}$$

Here, $\Delta\text{Health}(e)$ represents the decrease in health of enemy unit $e$ during a given timestep, and $\mathbb{I}(\cdot)$ is an indicator function. $\text{Reward}_{\text{death}}$ and $\text{Reward}_{\text{win}}$ are set to 10 and 200, respectively.

Table E.1: Detailed descriptions of SMAC-Hard+ scenarios

| Map | Ally Units | Enemy Units | State Dimension | Obs Dimension | Num. of Actions |
|---|---|---|---|---|---|
| 5m_vs_6m | 5 Marines | 6 Marines | 98 | 55 | 12 |
| MMM2 | 1 Medivac, 2 Marauders, 7 Marines | 1Medivac, 3 Marauders, 8 Marines | 322 | 176 | 18 |
| 27m_vs_30m | 27 Marines | 30 Marines | 1170 | 285 | 36 |
| Corridor | 6 Zealots | 24 Zerglings | 282 | 156 | 30 |
| 3s5z_vs_3s6z | 3 Stalkers, 5 Zealots | 3 Stalkers, 6 Zealots | 230 | 136 | 15 |
| 6h_vs_8z | 6 Hydralisks | 8 Zealots | 140 | 78 | 14 |

### E.2 GOOGLE RESEARCH FOOTBALL (GRF)

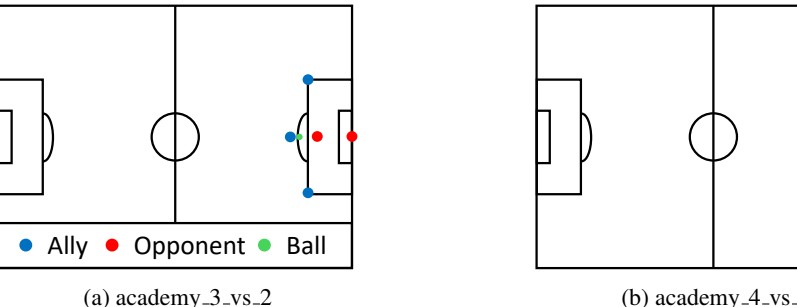

(a) academy_3_vs_2  (b) academy_4_vs_3

Figure E.2: Visualizations of GRF scenarios

GRF Kurach et al. (2020) offers a multi-agent soccer environment where each player is controlled by an autonomous agent. The game models realistic ball physics, player motion, and interaction mechanics such as tackling and passing. Teams must coordinate to achieve scoring opportunities while competing against opponents driven by scripted behaviors. From GRF scenarios, we consider `academy_3_vs_1_with_keeper` and `academy_4_vs_2_with_keeper` scenarios, which we abbreviate as `academy_3_vs_2` and `academy_4_vs_3` for brevity. Fig. E.2 illustrates the initial positions of the entities on the field, while Table E.2 summarizes the considered scenarios.

**State space**: The global state $s$ consists of all player positions and velocities, as well as ball position and velocity. The data for both ally and opposing teams are set to same.

**Observation space**: The observation for an agent includes local information about the ego player, nearby teammates, opponents, and ball-related features, all expressed relative to the agent's current frame.

**Action space**: The discrete action space of GRF covers movement in eight directions, sliding, passing, shooting, sprinting, and standing still, all of which are necessary in order to achieve scoring opportunities.

**Reward function** GRF provides two primary reward settings: Scoring and Checkpoint. Scoring function rewards agents with a +1 reward for scoring a goal and a -1 penalty for conceding one. While the Checkpoint function provides additional intermideate rewards. For example agents may receive rewards for successful passes or defensive actions. In our experiments, we follow the more sparse Scoring function, for more challenging scenarios.

Table E.2: Detailed description of GRF scenarios

| Scenario | Ally | Opponent | State Dim | Obs Dim | Action Dim |
|----------|------|----------|-----------|---------|------------|
| `academy_3_vs_2` | 3 central midfield | 1 goalkeeper, 1 center back | 26 | 26 | 19 |
| `academy_4_vs_3` | 4 central midfield | 1 goalkeeper, 2 center back | 34 | 34 | 19 |

### E.3   SMAC-COMM

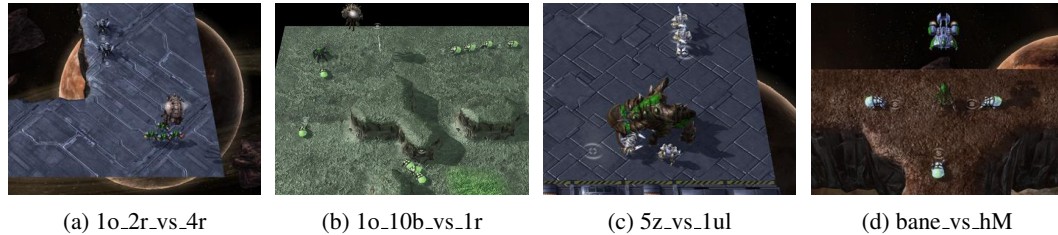

(a) 1o_2r_vs_4r      (b) 1o_10b_vs_1r      (c) 5z_vs_1ul      (d) bane_vs_hM

Figure E.3: Visualizations of SMAC-Comm scenarios

SMAC-Comm shares the same state, observation, action space, and reward function as the previously introduced SMAC-Hard+, but its tasks are specifically designed to emphasize communication. We illustrate the scenarios from SMAC-Comm considered in our experiments in Fig. E.3, and summarize them as Table E.3.

Table E.3: Detailed description of SMAC-Comm scenarios

| Map | Ally Units | Enemy Units | State Dimension | Obs Dimension | Num. of Actions |
|-----|-----------|-------------|-----------------|---------------|-----------------|
| 1o_2r_vs_4r | 1 Overseer, 2 Roaches | 4 Reapers | 68 | 49 | 10 |
| 1o_10b_vs_1r | 1 Overseer, 10 Banelings | 1 Roach | 148 | 84 | 7 |
| 5z_vs_1ul | 5 Zealots | 1 Ultralisk | 63 | 35 | 7 |
| bane_vs_hM | 3 Banelings | 1 Hydralisk, 1 Medivac | 52 | 35 | 8 |

## E.4 SMACv2

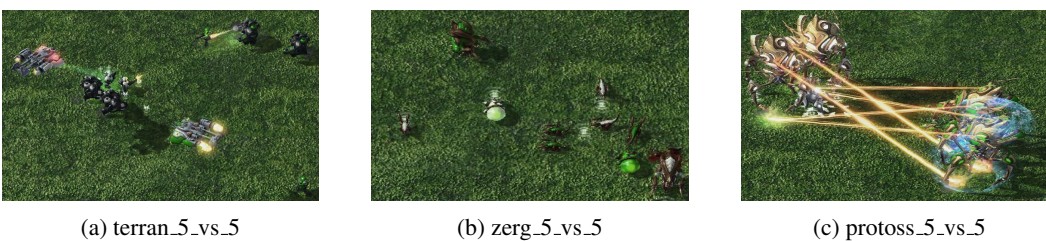

| (a) terran_5_vs_5 | (b) zerg_5_vs_5 | (c) protoss_5_vs_5 |

Figure E.4: Visualizations of SMACv2 scenarios

SMACv2 (Ellis et al., 2023) extends the original SMAC (Samvelyan et al., 2019) benchmark to provide a more rigorous testbed for evaluating generalizable cooperative multi-agent learning. While the state, observation, and action spaces, as well as the reward function, remain similar to SMAC, several key modifications distinguish SMACv2. First, instead of fixed map configurations, SMACv2 introduces randomized initializations (e.g., unit positions, health, numbers, and attributes), which prevent agents from overfitting to static scenarios. Second, it emphasizes greater unit diversity and asymmetric matchups, requiring more sophisticated tactical coordination among agents. Third, the difficulty scaling mechanism is refined: rather than varying only the number or placement of enemy units, environmental factors are also randomized, creating a natural train–test distribution gap. In our experiments, we consider the terran_5_vs_5, zerg_5_vs_5, protoss_5_vs_5 scenarios. Visualizations for each scenario are provided in Fig. E.4, and the scenario-specific details are summarized in Table E.4.

Table E.4: Detailed description of SMACv2 scenarios. Probabilities indicate the sampling distribution for randomized compositions.

| Map | Ally Units (prob.) | Enemy Units (prob.) | State Dim | Obs Dim | Num. Actions |
|---|---|---|---|---|---|
| terran_5_vs_5 | Marine (0.45)
Marauder (0.45)
Medivac (0.1) | Marine (0.45)
Marauder (0.45)
Medivac (0.1) | 120 | 82 | 11 |
| zerg_5_vs_5 | Zergling (0.45)
Hydralisk (0.45)
Baneling (0.1) | Zergling (0.45)
Hydralist (0.45)
Baneling(0.1) | 120 | 82 | 11 |
| protoss_5_vs_5 | Zealot (0.45)
Stalker (0.45)
Colossus (0.1) | Zealot (0.45)
Stalker (0.45)
Colossus (0.1) | 130 | 92 | 11 |

## E.5 SMAC-HARD (MIXTURE OPPONENT)

SMAC-Hard (Mixture Opponent) (Deng et al., 2024) shares the same state, observation, action space, and reward function as the previously introduced SMAC-Hard+, but it incorporates additional stochasticity by providing multiple opponent strategies. To distinguish it from the tasks from original SMAC-Hard+, we attach _Hard to each scenario to denote tasks from SMAC-Hard (Mixture Opponent). We summarize the scenarios we consider in our experiments as Table E.3.

Table E.5: Detailed description of SMAC-Hard (Mixture Opponent) scenarios

| Map | Ally Units | Enemy Units | State Dimension | Obs Dimension | Num. of Actions |
|---|---|---|---|---|---|
| 2s3z_Hard | 2 Stalkers,
3 Zealots | 2 Stalkers
3 Zealots | 120 | 80 | 11 |
| 3s_vs_5z_Hard | 3 Stalkers, | 5 Zealots | 68 | 48 | 11 |
| 5m_vs_6m_Hard | 5 Marines | 6 Marines | 98 | 55 | 12 |

# F    EXPERIMENTAL DETAILS

In this section, we provide the experimental details of the proposed S2Q. All experiments provided in this paper are conducted on a GPU server equipped with an NVIDIA GeForce RTX 3090 GPU and an Intel Xeon Gold 6348 (2.60GHz, 28 cores) processor, running Ubuntu 20.04. In Appendix F.1, we introduce popular CTDE baselines, including VDN, QMIX, and QPLEX. While we follow the original implementations and loss scaling of the prior baseline CTDE algorithms for shared hyper-parameters, we focus our parameter search on the hyperparameters newly introduced in S2Q. This hyperparameter setup is provided in Appendix F.2. Finally, Appendix F.3 details various MARL methods that address the limitations of monotonic value decomposition, as well as communication-focused MARL algorithms.

## F.1    DETAILS OF CONSIDERED CTDE BASELINES

**VDN** (Sunehag et al., 2017) is a cooperative MARL method based on value factorization, decomposing the joint action-value function into a sum of individual agent value functions, enabling centralized training with decentralized execution. The global $Q$-value is defined as:

$$Q^{\text{tot}}(s_t, \mathbf{a}_t) = \sum_{i=1}^{N} Q^i(\tau_t^i, a_t^i),$$

where individual $Q^i$ depends only on the agent's own trajectory $\tau_t^i$ and the chosen action $a_t^i$.

**QMIX** (Rashid et al., 2020a) introduces a more flexible, non-linear function for combining individual agent utilities into a joint action-value. To ensure that maximizing each agent's local $Q^i$ aligns with maximizing the global objective, QMIX enforces a monotonicity constraint between the joint value and the individual utilities, formalized as:

$$\frac{\partial Q^{\text{tot}}}{\partial Q^i} \geq 0, \quad \forall i,$$

**QPLEX** (Wang et al., 2020a) advances the expressiveness of value factorization by introducing a duplex dueling architecture while still satisfying the IGM (Individual-Global-Max) property. QPLEX decomposes the $Q$-values into advantage functions and formulates IGM as:

$$\arg \max_{\mathbf{a}} A^{\text{tot}}(\boldsymbol{\tau}, \mathbf{a}) = \begin{pmatrix} \arg \max_{a^1} A^1(\tau^1, a^1), \\ \vdots \\ \arg \max_{a^N} A^N(\tau^N, a^N) \end{pmatrix},$$

where $A^{\text{tot}}$ and $A^i$ represent the joint and individual advantage functions, respectively. The joint $Q$-value is then constructed as:

$$Q^{\text{tot}}(\boldsymbol{\tau}, \mathbf{a}) = \sum_{i=1}^{N} Q^i(\boldsymbol{\tau}, a^i) + \sum_{i=1}^{n} (\lambda_i(\boldsymbol{\tau}, \mathbf{a}) - 1) A^i(\boldsymbol{\tau}, a^i),$$

with the weighting factors $\lambda_i(\boldsymbol{\tau}, \mathbf{a}) > 0$ generated via a multi-head attention mechanism, enhancing the flexibility of the model.

Throughout our experiments, results for VDN, QMIX, and QPLEX are based on the implementations provided in PyMARL2 (Hu et al., 2021), an open-source MARL framework that includes implementations of diverse algorithms along with various improvements, such as TD($\lambda$), larger batch sizes, and the Adam optimizer. The code is available at: `https://github.com/hijkzzz/pymarl2`.

## F.2 HYPERPARAMETER SETUP

In our experiments, while we adopt the default hyperparameter configuration from PyMARL2 (Hu et al., 2021) for settings shared across CTDE methods, we focus on identifying effective values for the hyperparameters introduced in S2Q. The best parameters for each scenario are summarized in Table F.2. According to Table F.2, across a variety of tasks, $K = 2$, $T = 0.1$, and $\alpha = 1.0$ consistently achieve the best performance. While the weighting factor $w$ exhibits some task-specific variation, values of $0.75$ or $0.9$ generally perform well. These results demonstrate the robustness and flexibility of S2Q, achieving high win rates and rapid convergence across diverse scenarios.

Table F.1: Common $Q$-learning Hyperparameters

| Hyperparameter | Value |
|---|---|
| $\epsilon$ Decay Value | $1.0 \to 0.05$ |
| $\epsilon$ Anneal Time | 100000 |
| Target Update Interval | 200 |
| Discount Factor $\gamma$ | 0.99 |
| Buffer Size | 5000 |
| Batch Size | 128 |
| Learning Rate | 0.001 |
| Optimizer | Adam |
| Optimizer Alpha | 0.99 |
| Optimizer Eps | 1e-5 |
| Gradient Clip Norm | 10.0 |
| Num GRU Layers | 1 |
| RNN Hidden State Dim | 64 |
| Double Q | True |

Table F.2: Scenario-specific hyperparameter setup of S2Q

| Scenario | $K$ | $T$ | $\alpha$ | $w$ |
|---|---|---|---|---|
| **SMAC-Hard+** | | | | |
| 5m_vs_6m | 2 | 0.1 | 1.0 | 0.9 |
| MMM2 | 2 | 0.1 | 1.0 | 0.9 |
| 27m_vs_30m | 2 | 0.1 | 1.0 | 0.9 |
| corridor | 2 | 0.1 | 1.0 | 0.75 |
| 6h_vs_8z | 2 | 0.1 | 1.0 | 0.9 |
| 3s5z_vs_3s6z | 2 | 0.1 | 1.0 | 0.75 |
| **GRF** | | | | |
| academy_3_vs_2 | 2 | 0.1 | 1.0 | 0.75 |
| academy_4_vs_3 | 2 | 0.1 | 1.0 | 0.75 |
| **SMAC-Comm** | | | | |
| 1o_2r_vs_4r | 2 | 0.1 | 1.0 | 0.9 |
| 1o_10b_vs_1r | 2 | 0.1 | 1.0 | 0.9 |
| 5z_vs_1ul | 2 | 0.1 | 1.0 | 0.9 |
| bane_vs_hM | 2 | 0.1 | 1.0 | 0.9 |
| **SMACv2** | | | | |
| terran_5_vs_5 | 2 | 0.1 | 1.0 | 0.75 |
| zerg_5_vs_5 | 2 | 0.1 | 1.0 | 0.75 |
| protoss_5_vs_5 | 2 | 0.1 | 1.0 | 0.75 |
| **SMAC-Hard (Mixture Opponent)** | | | | |
| 5m_vs_6m_Hard | 2 | 0.1 | 1.0 | 0.9 |
| 2s3z_Hard | 2 | 0.1 | 1.0 | 0.9 |
| 3s_vs_5z_Hard | 2 | 0.1 | 1.0 | 0.9 |

### F.3 DESCRIPTION OF OTHER MARL METHODS FOR COMPARISON

**QMIX** (Rashid et al., 2020a): introduces a mixing network that combines per-agent value functions into a joint action-value under a monotonicity constraint, ensuring consistency between centralized training and decentralized execution. We base our implementation of QMIX on the following repository: `https://github.com/hijkzzz/pymarl2`

**WQMIX** (Rashid et al., 2020b): extends QMIX by introducing weighted projections in the mixing process, alleviating the representational limitations of monotonicity and enabling more accurate learning of optimal joint action-values. We base our implementation of WQMIX on the following repository: `https://github.com/hijkzzz/pymarl2`

**RiskQ** (Shen et al., 2023): introduces a quantile-based value factorization that models joint return distributions as weighted mixtures of per-agent utilities, satisfying the risk-sensitive IGM principle and enabling coordination under uncertainty. The official code can be found at: `https://github.com/xmu-rl-3dv/RiskQ`

**PAC** (Zhou et al., 2022): leverages counterfactual predictions of optimal joint actions to provide assistive information for value factorization, using a novel counterfactual loss and variational encoding to improve coordination under partial observability. The official code can be found at: `https://github.com/hanhanAnderson/PAC-MARL`

**FOP** (Zhang et al., 2021): factorizes the optimal joint policy in maximum-entropy MARL into individual actor-critic policies, with theoretical guarantees of convergence to the global optimum. The official code can be found at: `https://github.com/PKU-RL/FOP-DMAC-MACPF?tab=readme-ov-file`

**DOP** (Wang et al., 2020d): integrates value function decomposition into multi-agent actor-critic methods, enabling efficient off-policy learning while addressing credit assignment and centralized-decentralized mismatch, with guarantees of convergence. The official code can be found at: `https://github.com/TonghanWang/DOP`

**MARR** (Yang et al., 2024): improves sample efficiency in MARL by introducing a reset strategy and data augmentation, enabling high-replay training in parallel environments with fewer environment interactions. The official code can be found at: `https://github.com/CNDOTA/ICML24-MARR`

**MASIA** (Guan et al., 2022): enables efficient multi-agent communication by aggregating received messages into compact, task-relevant representations using a permutation-invariant encoder and self-supervised objectives, improving coordination and decision-making. The official code can be found at: `https://github.com/chenf-ai/Multi-Agent-Communication-Considering-Representation-Learning`

**NDQ** (Wang et al., 2019): combines value function factorization with communication minimization, enabling agents to act independently most of the time while selectively exchanging messages using information-theoretic regularizers to improve coordination. The official code can be found at: `https://github.com/TonghanWang/NDQ`

**MAIC** (Yuan et al., 2022): enables agents to generate targeted incentive messages that directly influence teammates' value functions, promoting efficient explicit coordination while remaining compatible with different value function factorization methods. The official code can be found at: `https://github.com/mansicer/MAIC`

**CAMA** (Shao et al., 2023): uses complementary attention to enhance high-contribution entities and compress low-contribution ones, addressing distracted attention and limited observability, thereby improving coordination in cooperative MARL. The official code can be found at: `https://github.com/thu-rllab/CAMA`

**T2MAC** (Sun et al., 2024): enables agents to communicate selectively with trusted partners and integrate information at the evidence level, improving coordination and communication efficiency in cooperative MARL. The official code can be found at: `https://github.com/ZangZehua/T2MAC`

# G  ADDITIONAL EXPERIMENT RESULTS

In this section, we present additional experiment results to further demonstrate the effectiveness of our proposed S2Q method. In Appendice G.1 and G.2, we provide additional performance evaluation on SMACv2 (Ellis et al., 2023) and SMAC-Hard (Mixture Opponent) (Deng et al., 2024), while we extend S2Q to other representative CTDE baselines, VDN and QPLEX, in Appendix G.3.

## G.1  PERFORMANCE COMPARISON IN SMACV2

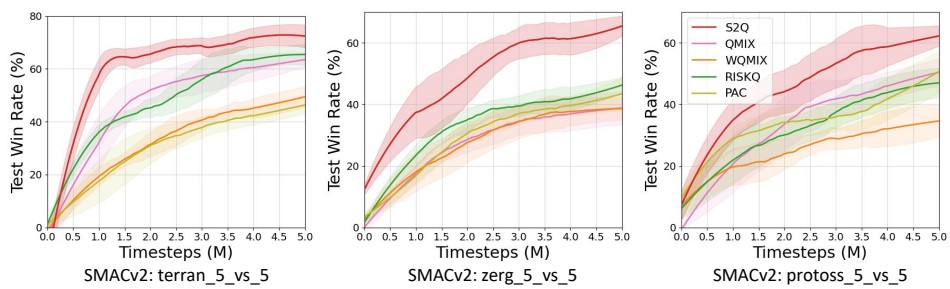

Figure G.1: Performance comparison: Average test win rates in the SMACv2 tasks

From Fig. G.1, which shows the performance in SMACv2 scenarios, we can observe that S2Q demonstrates superior performance over the MARL baselines across all scenarios, achieving both superior performance and faster convergence. Notably, its advantage is most pronounced in the zerg_5_vs_5 and protoss_5_vs_5 scenarios, where the high degree of stochasticity induced by varying ally and enemy team compositions makes the environment particularly challenging. In this setting, S2Q demonstrates its ability to effectively track the values of suboptimal actions, thereby enabling more efficient exploration and guiding the $Q^{\text{tot}}$ towards the optimal policy.

## G.2  PERFORMANCE COMPARISON IN SMAC-HARD (MIXTURE OPPONENT)

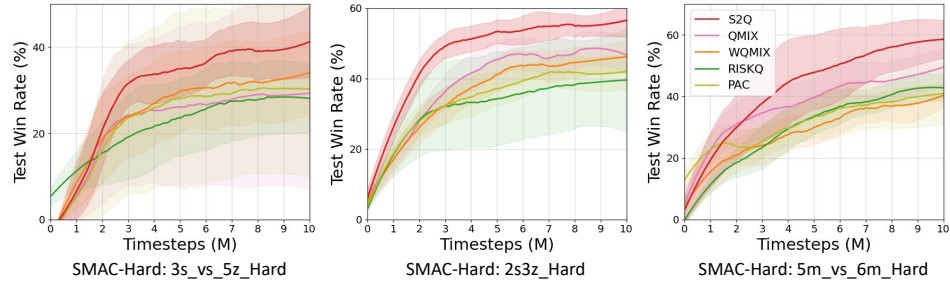

Figure G.2: Performance comparison: Average test win rates in the SMAC-Hard (Mixture Opponent) tasks

Fig. G.2 illustrates the performance on SMAC-Hard (Mixture Opponent) scenarios, including 5m_vs_6m_Hard, 2s3z_Hard, and 3s_vs_5z_Hard. As mentioned in Appendix E.5, we append "_Hard" to each scenario name to distinguish from the the scenarios from the SMAC-Hard+ benchmark. According to the results, S2Q consistently outperforms all MARL baselines across every considered task, demonstrating its superiority. Notably, in the 3s_vs_5z_Hard scenario, where agents must coordinate precise hit-and-run maneuvers against opponents that exhibit diverse tactical behaviors, S2Q demonstrates outstanding adaptability. This suggests that S2Q captures subtle opponent-conditioned value structures that conventional MARL baselines fail to exploit.

## G.3 GENERALITY OF S2Q ACROSS CTDE METHODS

Although our primary discussion of S2Q has been in the context of WQMIX, the proposed procedure is general and readily extends to other CTDE methods, such as VDN (Sunehag et al., 2017) and QPLEX (Wang et al., 2020a). By replacing the joint action-value function $Q^*$ with $Q^{\text{tot}}$, S2Q can be integrated into these value-decomposition baselines without modification. Across all scenarios, we observed significant performance improvements over the corresponding baselines. Consistent with the results obtained in WQMIX, the performance gains were most evident in the challenging 6h_vs_8z environment, which requires extensive exploration. These results demonstrate the practicality and robustness of S2Q, highlighting its ability to follow changes in the value landscape more closely and adapt to new optima faster than conventional approaches.

Table G.1: Performance comparison on SMACv2 environments

| Scenario | VDN | VDN+S2Q | QPLEX | QPLEX+S2Q |
|---|---|---|---|---|
| 5m_vs_6m | $62.47 \pm 11.12$ | $68.25 \pm 4.78$ | $58.18 \pm 3.52$ | $61.86 \pm 2.65$ |
| MMM2 | $0.00 \pm 0.00$ | $14.12 \pm 2.43$ | $50.78 \pm 10.63$ | $58.03 \pm 8.27$ |
| 27m_vs_30m | $13.75 \pm 5.33$ | $61.35 \pm 2.33$ | $47.34 \pm 8.26$ | $63.18 \pm 5.54$ |
| Corridor | $56.25 \pm 26.14$ | $70.49 \pm 13.37$ | $48.62 \pm 24.02$ | $64.68 \pm 18.51$ |
| 6h_vs_8z | $8.94 \pm 4.31$ | $48.03 \pm 9.37$ | $3.44 \pm 1.13$ | $41.32 \pm 6.23$ |
| 3s5z_vs_3s6z | $47.66 \pm 19.66$ | $49.82 \pm 10.31$ | $43.19 \pm 17.02$ | $48.73 \pm 10.46$ |

# H    ADDITIONAL ABLATION STUDY

In this section, we provide a deeper analysis on S2Q. In Appendix H.1, we provide further analysis on the hyperparameters $K$ and $T$ in additional tasks, while in Appendix H.2, provide additional ablation study on the suppression constant $\alpha$ and weighting factor $w_c$. Furthermore, in Appendix H.3, we analyze the role of auxiliary signals in scenarios where communication is not critical, and in Appendix H.4, provide additional examples of shift in value function along with trajectory analysis in SMAC-Comm environment. Finally, Appendix H.5 presents computational complexity analysis on S2Q, to show that the computational overhead from leveraging multiple sub-value functions remains moderate, compared to the significant performance gain of S2Q.

## H.1    HYPERPARAMETER ANALYSIS IN ADDITIONAL TASKS

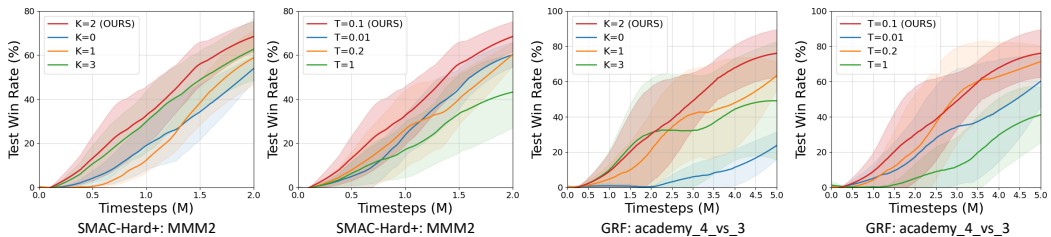

Figure H.1: Hyperparameter analysis. From left to right: (1) effect of $K$ in MMM2, (2) effect of temperature $T$ in MMM2, (3) effect $K$ in academy_4_vs_3, and (4) effect of $T$ in academy_4_vs_3

To further emphasize the generality of S2Q beyond the settings examined in Section 5.3, we additionally evaluate the effects of the number of sub-value networks $K$ and the Softmax temperature $T$ in two more challenging environments: MMM2 scenario from SMAC-Hard+ and academy_4_vs_3 scenario from GRF.

**Number of sub-value functions $K$:** As in earlier results, $K = 2$ achieves the best overall performance in both MMM2 and academy_4_vs_3, offering the strongest balance between convergence speed and final win rate. In contrast, $K = 0$ fails to provide meaningful exploratory diversity, while larger values such as $K = 3$ introduce increased variance and mild instability. These observations further support that a moderate number of candidate sub-networks yields a favorable trade-off between diversity and stability.

**Softmax temperature $T$:** A moderate temperature (e.g., $T = 0.1$) provides the most effective balance between focused sampling and sufficient exploration. Very low temperature values ($T = 0.01$) result in overly deterministic behavior early in training, suppressing useful exploration, while higher temperatures (e.g., $T = 1.0$) promote excessive exploration and slow convergence. Notably, $T = 0.2$ performs competitively, indicating that S2Q remains robust to reasonable variations in temperature as long as sampling remains sufficiently concentrated on promising sub-values.

Across both environments, we observe trends that closely mirror those reported earlier for 6h_vs_8z, demonstrating that the behavior of S2Q is consistent and robust across domains. Overall, the additional experiments on SMAC-Hard+ and GRF confirm that S2Q maintains stable and consistent performance characteristics across diverse and challenging multi-agent environments, reinforcing its generality and robustness.

## H.2 ABLATION STUDY ON ADDITIONAL HYPERPARAMETERS

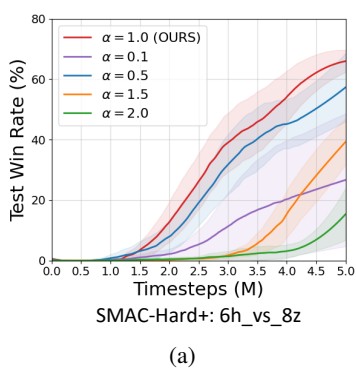
(a)

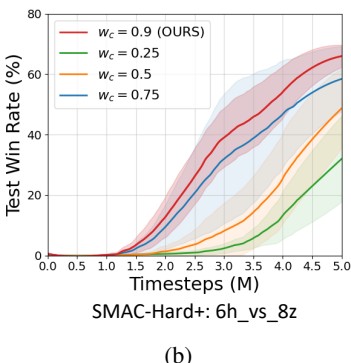
(b)

Figure H.2: Hyperparameter analysis. Effect of (a) suppression constant $\alpha$ and (b) weight factor $w_c$.

**Suppression constant** $\alpha$: Fig. H.2(a) evaluates the effect of the suppression constant $\alpha \in [0.1, 0.5, 1.0, 1.5, 2.0]$, which determines how strongly the values of previously identified actions are reduced and thereby controls how far sampled suboptimal actions lie from the current optimum. The results indicate that $\alpha = 1.0$ yields the best performance: this setting encourages exploration of meaningful, moderately distant sub-actions that are informative for updating the value landscape. A smaller value, $\alpha = 0.5$, also performs well since it biases sampling toward nearby sub-actions that remain relevant to the optimum. In contrast, a very small value such as $\alpha = 0.1$ fails to sufficiently suppress the previously selected optimal actions, preventing the discovery of meaningful suboptimal actions and resulting in a significant degradation in performance. On the other hand, larger values ($\alpha = 1.5, 2.0$) substantially hurt performance: excessive suppression forces sampling of actions that are very distant from the optimum, producing updates that are less informative (and often misleading) for tracking the true optimal policy, which destabilizes learning and slows convergence. These findings suggest that a moderate suppression level is necessary to balance exploration breadth with the relevance of sampled sub-actions.

**Weighting factor** $w_c$: Fig. H.2(b) reports the effect of varying the weighting factor $w_c$, which determines how TD-error is scaled by reinforcing corrective signals that reduce underestimation. The results show that $w_c = 0.9$ yields the highest win rates, striking a balance between effective error correction and stable value learning. A slightly smaller value, $w_c = 0.75$, also performs well, indicating robustness to moderate relaxations. However, more aggressive reductions such as $w_c = 0.5$ and $w_c = 0.25$ significantly weaken the propagation of TD-error, leading to information loss and degraded performance. These findings highlight that maintaining a relatively high $w$ ensures that informative TD signals are consistently transmitted, while still guiding the value functions towards the optimal values.

## H.3 EFFECT OF AUXILIARY SUPERVISION

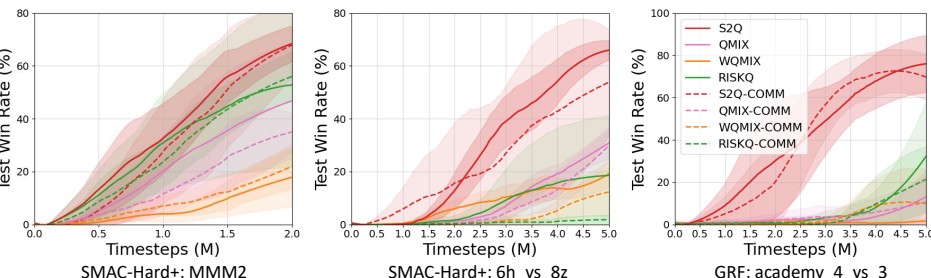

Figure H.3: Performance comparison: Average test win rates in the SMAC-Hard+ and GRF task for MARL algorithms and their communication variant

To demonstrate that the performance gains of S2Q arise from the proposed successive sub-value learning, rather than the auxiliary latent information, we conduct additional experiments on SMAC-Hard+ scenarios, including `MMM2` and `6h_vs_8z`) and `academy_4_vs_3` scenario from GRF. In these environments, we equipped all non-communication baselines with the same encoder-decoder architecture used in S2Q-Comm, and denote these variants by appending "`-Comm`". This setup ensures that every algorithm receives an identical latent signal derived from reconstructing the global state, thereby isolating the effect of the auxiliary representations from the effect of S2Q's successive sub-value learning.

The results, presented in Fig. H.3 show that injecting these latent-based auxiliary signals has minimal influence on performance across the considered tasks where communication is not critical for task success. In some cases, performance even degrades due to the latent introducing additional noise. Similar trends appear in the GRF `academy_4_vs_3` environment, where communication is not the primary bottleneck. Notably, S2Q-Comm consistently outperforms all baselines, including their "-Comm" variants. These findings confirm that the advantage of S2Q-Comm does not originate from the auxiliary state-reconstruction signal or the latent variable itself. Instead, the gains stem from S2Q's core mechanisms.

## H.4 SHIFT IN VALUE FUNCTION AND ADDITIONAL TRAJECTORY ANALYSIS

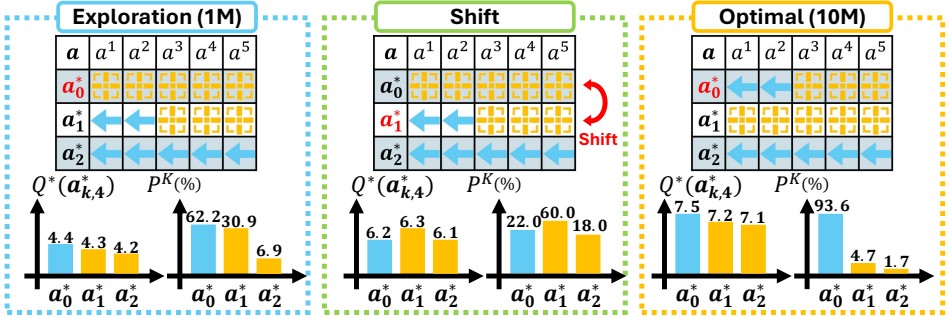

Figure H.4: Value function shift in `5m_vs_6m_Hard`)

In addition to the motivation discussed in Section 4.1 and the trajectory analysis presented in Section 5.2, we provide further examples of shifts in the value function. Specifically, we examine value-function changes in SMAC-Hard (Mixture Opponent) `5m_vs_6m_Hard` and GRF `academy_4_vs_3`, with results depicted in Fig. H.4 and H.5. In `5m_vs_6m_Hard`, agents initially focus on greedy attacks to secure easy eliminations, but the value function subsequently learns that repositioning to form favorable formations yields higher long-term returns, causing the optimal action to shift from *attack* to *move*.

Similarly, in `academy_4_vs_3`, agents first attempt low-success long shots, but eventually discover that approaching the goal or passing increases scoring probability, shifting the optimal action from

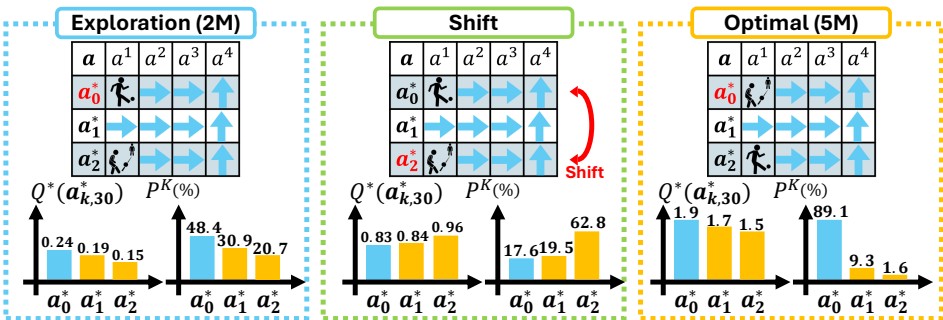

Figure H.5: Value function shift in `academy_4_vs_3`

*shoot* to *pass*. These examples further highlight S2Q's capability to track and adapt to evolving optimal behaviors across diverse environments.

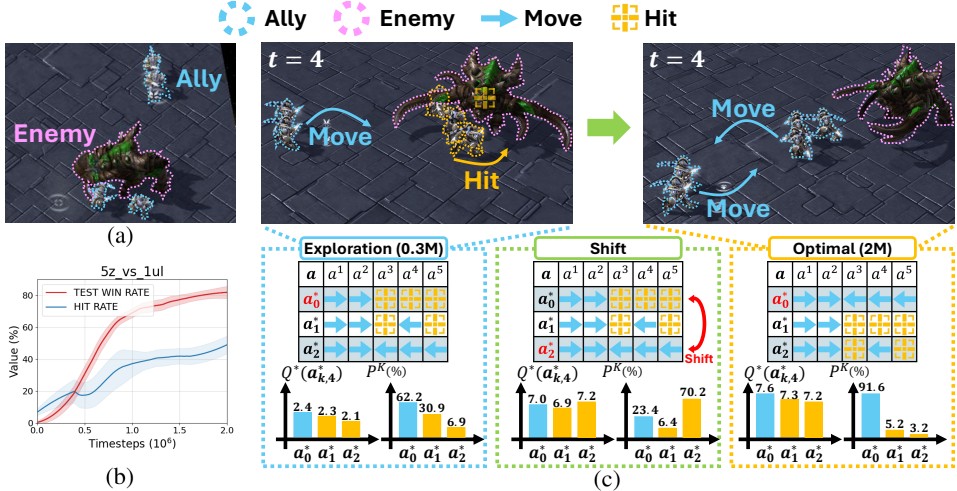

Figure H.6: Training behavior of S2Q. Left: training step = 0.3M. Right: training step = 2M

In addition, we further investigate the training dynamics of S2Q ($K = 2$), we analyze the `5z_vs_1ul` scenario from SMAC-Comm, where the agents must learn to coordinate based on the latent information $z$. `6h_vs_8z`. Unlike scenarios that reward aggressive engagement, the optimal strategy in `5z_vs_1ul` is to *move* and group with the distant ally, while direct *attack* actions often lead to heavy losses due to the enemy's superior strength. As shown in Fig.H.6(a), the action distribution reveals that agents initially favor *attack*, reflecting a local optimum that prioritizes immediate engagement. Consequently, the attack ratio increases in the early stages of training. However, as training progresses, S2Q enables agents to recognize the risks of premature aggression, resulting in a gradual decline in attack frequency and a corresponding rise in the use of *move*. Fig.H.6(b) illustrates how this shift is accompanied by improvements in survival time and overall win rate.

Fig. H.6(c) highlights how S2Q adapts to this changing notion of optimality. Early in training, *attack* is tracked as the dominant action ($\mathbf{a}^*_{0,t}$), while *move* is considered suboptimal. As the value landscape evolves, $Q^*$ reassigns higher returns to *move*, prompting S2Q to increase its execution frequency through the Softmax-based behavior policy. This dynamic reallocation allows $Q^{\text{sub}}_0$ to align with the true optimal strategy, thereby reducing reliance on local optima. These findings emphasize that even in environments requiring implicit coordination and communication, such as `5z_vs_1ul`, S2Q effectively leverages its successive sub-value component to explore, track, and ultimately converge to the optimal policy.

## H.5 COMPUTATIONAL COMPLEXITY

S2Q introduces additional computational overhead due to its use of multiple sub-value functions and the encoder-decoder architecture for coordinating $k$-selection. To evaluate this overhead and demonstrate that it remains minimal relative to performance benefits, we compare S2Q and QMIX in controlled experiments on an NVIDIA GeForce RTX 3090 GPU and an Intel Xeon Gold 6348 (2.60GHz, 28 cores) processor running Ubuntu 20.04 with PyTorch. In `MMM2` and `27m_vs_30m` scenarios where QMIX achieves competitive performance, we measure the time required to reach a 50% win rate. Table H.1 summarizes the results. According to Table H.1, S2Q incurs only a moderate increase in computation, approximately 8.4% for `MMM2` and 3.5% for `27m_vs_30m`. More importantly, due to faster learning enabled by successive sub-value function tracking, S2Q reaches the target win rate of 50% significantly faster than QMIX, demonstrating that the modest overhead is outweighed by the improved convergence speed and overall performance.

Table H.1: Comparison of training time between S2Q (for different $K$) and QMIX.

| Scenario | Metric | S2Q (K=1) | S2Q (K=2) | S2Q (K=3) | QMIX |
|---|---|---|---|---|---|
| `MMM2` | Time / 1M | 122.0 min | 129.3 min | 136.5 min | 119.3 min |
| | $T$ at 50% win | 1.48M | 1.28M | 1.67M | 2.16M |
| | Time at 50% win | 180.6 min | 165.5 min | 228.0 min | 257.7 min |
| `27m_vs_30m` | Time / 1M | 241.0 min | 245.1 min | 258.0 min | 236.8 min |
| | $T$ at 50% win | 1.39M | 1.15M | 1.57M | 1.86M |
| | Time at 50% win | 335.0 min | 281.9 min | 405.1 min | 440.4 min |

In addition to time-wise comparison, we also evaluate GPU memory consumption to quantify the overhead introduced by maintaining multiple sub-value networks. Table H.2 reports peak GPU usage for S2Q with different values of $K$ and for QMIX. According to the results, memory usage increases moderately with $K$ due to the additional sub-networks, but the overall overhead remains reasonable: in `MMM2`, S2Q with $K = 2$ requires only 248 MB more memory than QMIX, and even $K = 3$ remains within a 19% increase. A similar pattern holds in the more computationally demanding `27m_vs_30m` scenario, where S2Q($K = 2$) uses 336 MB more memory than QMIX. The incremental cost from $K = 2$ to $K = 3$ is also modest, considering the added representational capacity. Importantly, this memory overhead scales predictably and linearly with $K$.

Combined with the substantial reduction in training time reported in Table H.1, these results demonstrate that S2Q achieves significant improvements in learning efficiency while incurring only limited computational and memory overhead. This confirms that the successive sub-value learning framework offers a favorable trade-off between memory usage and performance gains, even in large-scale SMAC scenarios.

Table H.2: Comparison of memory usage between S2Q (for different $K$) and QMIX.

| Scenario | Metric | S2Q (K=1) | S2Q (K=2) | S2Q (K=3) | QMIX |
|---|---|---|---|---|---|
| `MMM2` | Perk GPU usage (MB) | 1967.0 MB | 2087.0 MB | 2188.0 MB | 1839.0 MB |
| `27m_vs_30m` | Perk GPU usage (MB) | 3665.0 MB | 3761.0 MB | 4076.0 MB | 3425.0 MB |

