# OpenReview forum: "Retaining Suboptimal Actions to Follow Shifting Optima in Multi-Agent Reinforcement Learning"
_ICLR.cc/2026/Conference — ICLR 2026 Poster_

### Official Review · Reviewer_44v8 · 2025-10-30

**Soundness:** 3
**Presentation:** 3
**Contribution:** 2
**Rating:** 4
**Confidence:** 5

**Summary:**

This paper proposes S2Q, a new framework for cooperative MARL under CTDE settings. This paper retains and learns from suboptimal actions so that the system can better adapt when the optimal joint policy changes during the training. Instead of relying on single best action, S2Q successively trains multiple sub-value functions that each caputure distinct high-value joint actions. By combining them through softmax-based behaviour policy, the method encourages persistent exploration and rapid adjustment to new optima. Experiments on SMAC, SMAC-Comm, and GRF shows outperformance to baseline algorithm.

**Strengths:**

1. Originality: The paper presents an original idea of retaining suboptimal actions through the proposed S2Q framework. This introduces a new perspective in value decomposition MARL by modeling multiple high-value modes instead of a single optimal action, addressing a long-standing issue of adaptability under shifting optima.
2. Quality: The technical formulation is solid and well-motivated, supported by a clear derivation of objectives and loss functions. The experimental setup is comprehensive, evaluating the method on multiple challenging benchmarks and performing ablations to validate each component.
3. Clarity: The paper is clearly written, with well-organized sections, intuitive figures, and a motivating example (the payoff matrix) that effectively conveys the main idea. The methodology and algorithm are clearly explained and reproducible, with good visual aids to guide understanding.
4. Significance: The contribution is significant for cooperative MARL. It addresses a key limitation in existing value decomposition approaches, poor adaptability when optimal actions shift. The framework’s generality and empirical robustness indicate potential impact on future MARL research.

**Weaknesses:**

1. Limited theoretical depth. While the paper provides a solid conceptual and empirical contribution, the theoretical analysis of why successive sub-value learning guarantees adaptability or improved convergence remains shallow. The paper lacks formal results on convergence properties, stability, or value approximation bounds for S2Q, which would strengthen its scientific aspect.
2. Incremental novelty relative to prior frameworks. Although the idea of retaining suboptimal actions is interesting, the implementation may be viewed as an extension of WQMIX with auxiliary sub-networks and a Softmax exploration mechanism. The paper could better clarify how S2Q fundamentally differs from or improves upon multi-head or ensemble-based exploration methods beyond heuristic layering.
3. Computational efficiency discussion. The paper acknowledges increased computation and memory overhead but only qualitatively. A quantitative analysis comparing GPU memory, or training wall-clock time to QMIX/WQMIX would make the practicality argument more convincing, especially for scaling to large agent teams.
4. Clarity of communication variant. The role of the encoder–decoder communication module is somewhat underexplained. It is unclear how much of the improvement in SMAC-Comm tasks comes from the communication architecture itself versus the S2Q mechanism. A clearer disentanglement or comparison to standalone communication baselines would improve interpretability.
5. Missing experiment details. Although the codebases are described in the Appendix individually, it seems that the performance in Figure 5 of the QMIX algorithm originated from pymarl2 does not achieve the performance stated in RIIT[1]. A detailed training parameters or any modifications are better to be discussed.

[1] Rethinking the Implementation Tricks and Monotonicity Constraint in Cooperative Multi-Agent Reinforcement Learning

**Questions:**

Based on the weaknesses described above, I have some questions for the authors.

1. Could the authors provide a more formal justification or theoretical insight into why maintaining multiple sub-value functions helps track shifting optima more effectively?
2. Are there any guarantees on the convergence or stability of S2Q, particularly when sub-value suppression is applied iteratively? A sketch of theoretical grounding could greatly strengthen the contribution.
3. Could the authors provide a direct comparison or at least a discussion to clarify whether S2Q’s performance gains come from retaining suboptimal modes or simply from having additional representational capacity?
4. The paper mentions that the computational overhead is “moderate.” Could the authors quantify this or training time ratio relative to QMIX or WQMIX?
5. How does the overhead scale with the number of sub-value networks K and the number of agents N? This would help assess the method’s practicality in larger-scale MARL tasks.
6. Have the authors considered testing in deliberately non-stationary environments, such as mixed multiple opponent strategies in SMAC-Hard environment,to validate S2Q’s adaptability?
7. In SMAC-Comm results, how much improvement arises from the encoder–decoder communication mechanism versus the S2Q framework itself?
8. The paper positions S2Q as addressing the issue of “shifting optima.” Could the authors better relate this to other known issues such as non-stationarity, policy co-adaptation, or value landscape drift in MARL? Clarifying this link may help highlight the conceptual novelty and general applicability of S2Q.

If all the concerns are well-addressed, I would consider raising the score. Thank you.

---

> ### Author Response · Authors · 2025-11-20
> **Response to Reviewer 44v8 (1/2)**
>
> ## Reviewer 44v8
> To thoroughly address the reviewer's comments and concerns, we have divided our rebuttal into two separate comments for clarity. We kindly ask for your understanding. In the first comment, we address the theoretical analysis of successive Q-learning and the difference between S2Q and previous methods. In the second comment, provide further analysis of S2Q with additional experiments, while clarifying the contribution of the communication variant and the concept of *shifting optima*. We sincerely appreciate your understanding and consideration.
>
> ---
>
> **Weakness 1/Question 2 (Theoretical analysis of Successive Q-learning):**
> We appreciate the reviewer’s insightful comments regarding the need for stronger theoretical support. To formally demonstrate that S2Q exactly identifies the $k$-th suboptimal action, we have added Theorem 4.1 in Section 4.2, together with a full proof. The theorem establishes that, under bounded rewards and with a sufficiently large suppression factor $\alpha$, each sub-value function $Q^{\mathrm{sub}}_k$ converges to the $k$-th successive suboptimal joint action of the underlying value function $Q^\ast$. In other words, minimizing the S2Q objective provably extracts the top-$k$ modes of $Q^*$ in a sequential and well-defined manner. Consequently, increasing $K$ does not introduce instability, since each sub-value function is guaranteed to converge to its designated level in the hierarchy, maintaining a consistent and ordered structure as $K$ grows.
>
> To ensure full rigor in the proof, we slightly modified the suppression term in Equation (2), replacing $Q_{\mathrm{targ}}^\ast$ with $\max(Q_{\mathrm{targ}}^\ast, C)$, where $C>0$ is a constant introduced to properly handle potentially negative values of $Q^\ast$. This modification is used only for the theoretical analysis. In all environments considered in our experiments, $Q^*$ quickly becomes positive during training, so the implementation remains unchanged and directly uses $Q^\ast$ without any additional modifications. We believe this theoretical development substantially strengthens the foundation and clarity of our method. We sincerely thank the reviewer for encouraging us to include this analysis.
>
> ---
>
> **Weakness 2/Question 1 (Novelty and Difference from ensemble methods):**
>
> Value-based MARL methods such as QMIX rely on the fact that Q-learning achieves global convergence only when the agent sufficiently explores diverse state–action pairs [R.1]. However, the IGM constraint prevents QMIX from accurately learning the values of non-maximal joint actions, limiting its ability to maintain meaningful exploration. S2Q addresses this issue not by generating arbitrary or population-based diversity but by using the global value function $Q^*$ to identify and prioritize true successive suboptimal actions. These actions are precisely those most likely to become optimal as the value landscape shifts during training. While QMIX and WQMIX depend on per-agent $\epsilon$-greedy exploration, whose coordinated deviations become exponentially unlikely with more agents, S2Q employs a softmax-based selection grounded directly in suboptimality. This yields more reliable visitation of informative actions and facilitates convergence in settings where structured exploration is critical. The matrix game analysis in Appendix D further illustrates how $\epsilon$-greedy collapses to the current optimal action, whereas S2Q continues to explore strategically valuable suboptimal modes.
>
> The distinction between S2Q and ensemble or multi-head approaches is also fundamental. Ensemble-style methods or multi-head architectures introduce multiple Q-heads to increase diversity, enhance representational capacity, or reduce variance, yet these heads are not constructed to correspond to globally consistent suboptimal actions and therefore do not form an ordered representation of the value landscape. In contrast, **S2Q explicitly learns sub-value functions that sequentially capture true successive suboptimal joint actions** derived from the underlying global value function. The objective is not merely to diversify Q-functions but to preserve and exploit precise suboptimal modes that are most likely to become optimal as learning progresses. Unlike ensemble or multi-head designs whose diversity is not tied to the structure of $Q^*$, S2Q leverages principled, value-grounded suboptimality to support structured exploration and adaptive policy improvement in large joint action spaces.
>
> [R.1]: Sutton, Richard S., and Andrew G. Barto. *Reinforcement learning: An introduction.* Vol. 1. No. 1. Cambridge: MIT press, 1998.

---

> ### Author Response · Authors · 2025-11-20
> **Response to Reviewer 44v8 (2/2)**
>
> **Weakness 3, Question 4, 5 (Computational cost analysis)**
> We thank the reviewer for highlighting the importance of computational overhead. Following the requests of the reviewer, we provide full quantitative comparisons of training time and GPU memory usage for S2Q in Appendix H.5. As we discuss in detail in our response to Reviewer xg2N, the computational overhead introduced by S2Q remains modest and scales reliably even in large-agent settings such as `27m_vs_30m`. We kindly refer the reviewer to our detailed responses to reviewer xg2N, where these additional analyses are fully presented.
>
> ---
>
> **Weakness 4/Question 7 (Clarity of communication variant)**
> As discussed in Section 4.3, S2Q-Comm injects a latent representation of the joint state into the value network. Therefore, if we remove the successive sub-value mechanism from S2Q, the communication variant effectively reduces to a standard state-reconstruction-based communication method. To validate this, we updated the comparison in the SMAC-Comm experiments introduced in Section 5.1 by adding an additional variant, QMIX-COMM, which augments standard QMIX with a state-reconstruction latent $z$. We have revised Fig. 6 to match the updated result, which shows a substantial performance gap, demonstrating that the **performance improvements in SMAC-Comm tasks are primarily attributable to the successive sub-value mechanism.**
>
> ---
>
> **Weakness 5 (Performance of fine-tuned QMIX)**
> RIIT [1] introduces a fine-tuned QMIX implementation trained for 10M steps, whereas our study employs shorter training horizons to evaluate convergence efficiency. This difference in training budgets naturally leads to discrepancies in reported performance. Importantly, the QMIX results we present are consistent with values commonly reported in recent MARL works that also adopt the RIIT implementation under comparable training settings [R.2, R.3]. In other words, the performance levels observed in our experiments align with what the broader literature typically reports when using similar default hyperparameters and training durations.
>
> [R.2] Shen, Siqi, et al. “RiskQ: risk-sensitive multi-agent reinforcement learning value factorization.” *NeurIPS 2023*
>
> [R.3] Yang, Yaodong, Guangyong Chen, and Pheng-Ann Heng. “Sample-efficient multiagent reinforcement learning with reset replay.” *ICML 2024*
>
> ---
>
> **Question 6 (Evaluation in additional benchmarks):**
> Thank you for your suggestion. As discussed in Section 5.1, in addition to our evaluation on the original SMAC environments, we have also conducted experiments on SMACv2 [R.4], a recently proposed benchmark that incorporates additional stochasticity through randomized team compositions and initial positions. The performance comparison in SMACv2 tasks is provided in Appendix G.1. As shown in Fig. G.1, S2Q consistently outperforms other MARL baselines across all considered scenarios: `terran_5_vs_5`, `zerg_5_vs_5`, and `protoss_5_vs_5`. These results demonstrate that S2Q is effective not only in the standard SMAC setting but also in newer, more stochastic, and more realistic benchmarks.
>
> Furthermore, following the reviewer’s recommendation, we evaluated S2Q on the recently proposed SMAC-Hard benchmark, where opponents adopt mixed scripted strategies to introduce higher levels of stochasticity and non-stationarity. To distinguish it from the original SMAC environment, we append “`_Hard`” to each scenario name, including `5m_vs_6m_Hard`, `2s3z_Hard`, and `3s_vs_5z_Hard`. The results, provided in Appendix G.2 and summarized in Fig. G.2, show that S2Q again achieves consistently superior performance over all baselines. These findings confirm that S2Q maintains strong and stable performance even in newly proposed, highly stochastic MARL environments.
>
> [R.4] Ellis, Benjamin, et al. “SMACv2: An improved benchmark for cooperative multi-agent reinforcement learning.” *NeurIPS 2023*.
>
> ---
>
> **Question 8 (Shifting optima and value landscape drift)**
> The phenomenon we address can be understood as value landscape drift: as agents learn, opponents adapt, or episode initial conditions vary (e.g., randomized unit placements or mixed opponent strategies), the mapping from joint actions to expected returns changes, causing the identity of the optimal joint action to shift over time. In our experiments, including the payoff game in Section 4.1 and trajectory analyses shown in Fig. 8 of Section 5.2, we observe this effect: the true optimal action changes across contexts, and S2Q’s maintained suboptimal candidates provide ready alternatives that can be promoted when the optimum shifts. This illustrates why explicitly tracking suboptimal actions, as S2Q does, is crucial for quickly adapting to changing value landscapes.
>
> ---
>
> We thank the reviewer once again for their thoughtful feedback and careful evaluation. We trust that the clarifications and additional analyses provided in our response address the reviewer's concerns.

---

> > ### Comment · Reviewer_44v8 · 2025-11-25
> >
> > I appreciate the authors’ detailed and thorough rebuttal. Most of my earlier concerns regarding clarity, presentation, and experimental evidence have been addressed. Although I am still questioning the theoretical contribution of selecting the suboptimal Q value in MARL, the additional explanations and newly provided empirical experiment results improve the quality and readability of the manuscript. In light of these revisions and clarifications, I am inclined to raise my score to 6.

---

> > > ### Author Response · Authors · 2025-11-25
> > >
> > > Dear reviwer 44v8
> > >
> > > We sincerely thank the reviewer for the positive feedback and for raising the score. Regarding the theoretical aspect, our primary goal is to clarify that S2Q can exactly select suboptimal joint actions under $Q^*$, in contrast to QMIX-based methods, which cannot select such actions due to the IGM constraint. Concerning the faster convergence behavior, we totally agree with the reviewer’s comment, but we address this point through extensive empirical analyses rather than a theoretical derivation, which is highly challenging in nonstationary MARL. We are very grateful for the reviewer’s thoughtful reassessment and constructive comments, which have helped us improve the clarity and presentation of our work.

---

### Official Review · Reviewer_xg2N · 2025-10-31

**Soundness:** 3
**Presentation:** 3
**Contribution:** 3
**Rating:** 8
**Confidence:** 4

**Summary:**

This paper proposes a method to overcome value function shifts during training in CTDE MARL, named S2Q. The proposed method learns a set of several sub-value functions, where each aims to identify different suboptimal actions.

**Strengths:**

This paper is well written and provides a very extensive set of experiments and ablations. The results are consistently strong across very diverse environments. While apparently simple, i find clever the idea to surpress the optimal actions in the calculations of subsequent value functions and the performances show big improvements in a range of tasks.

**Weaknesses:**

The authors could have provided a deeper analysis of the scalability of the proposed method, since it requires sequential computations using sub-networks. I.e, since there is a mixing for each Q, up until what point can k scale? In the communication encoder-decoder module in figure 3, the authors could have provided a better description of the architecture of these modules.

Please find below some more specific questions.

**Questions:**

1. Could the authors describe another more practical scenario aside from the provided matrix games where the optimal action shift might happen? For example in the experimented environments such as SMAC, when could such shift happen?
2. i find interesting the sudden boost around 1M timesteps in figure 6 for the task bane_vs_hM; have the authors explored why this is seen only in this specific environment, instead of a more linearly increasing curve?
3. how can you find the best k? is there any guarantees (theoretical maybe) that for higher k the variance and learning instability will increase (as mentioned in lines 454-455)?
4. considering the sub value functions are learned in a sequential manner, how does the proposed method relate to the higher order q-learning approach presented in [1]?

[1] https://arxiv.org/abs/2304.13383

---

> ### Author Response · Authors · 2025-11-20
> **Response to Reviewer xg2N**
>
> ### Reviewer xg2N
> We sincerely appreciate the reviewer’s thoughtful feedback. To address the raised concerns, we offer additional clarification regarding the scalability and core mechanism of S2Q.
>
> ---
>
> **Weakness 1/Question 3 (Scalability of $K$):**
> For theoretical completeness, we added Theorem 4.1 in Section 4.2, with a full proof. The theorem shows that, under bounded rewards and a sufficiently large suppression factor $\alpha$, each sub-value function $Q^{\mathrm{sub}}_k$ converges to the $k$-th suboptimal joint action of $Q^\ast$. In other words, minimizing the S2Q objective provably extracts the top-$k$ modes of $Q^\ast$ in a well-defined manner. Consequently, *increasing $K$ does not introduce instability*, since each sub-value function is guaranteed to converge to its designated level in the hierarchy.
>
> In practice, increasing $K$ introduces additional computational cost. We quantify this cost, measuring training time and GPU  usage of S2Q for different values of $K$, and compare them with QMIX. The results in Tables H.1 and H.2 of Appendix H.5 show that each sub-network increases training time and memory consumption by approximately $6\\%$ and $7\\%$, respectively, which we consider acceptable given the corresponding performance gains. However, as shown in Fig. 8, increasing $K$ is not always beneficial, since excessively poor suboptimal actions contribute little to learning while still incurring additional computational cost. In practice, $K=2$ offers a favorable balance, introducing modest overhead while achieving strong performance.
>
> ---
>
> **Weakness 2 (Detailed description of Fig. 3):**
> To clarify Fig. 3, we describe how S2Q uses local histories through two parallel paths: the S2Q learning path and the encoder–decoder path. In the S2Q learning path, each agent processes its own history $\tau_t^i$ to generate decentralized actions and the corresponding softmax probability $\mathbf{P}_t$. In the encoder–decoder path, all agents’ histories are aggregated, encoded into a latent vector, and decoded to predict the selection distribution $\hat{\mathbf{P}}_t$, which is trained to approximate $\mathbf{P}_t$. This estimated distribution is then used during **training only** to synchronize agents to select the same sub-value function $Q_k^{\text{sub}}$. We have updated Section 4.3 to incorporate this clarified explanation of Fig. 3.
>
> ---
>
> **Question 1 (Change in value function):**
> In Fig.7, we illustrated how the value function evolves using the `6h_vs_8z` scenario from SMAC. Early in training, the learned value function assigns a higher value to evasive behaviors, as the ally agents are outnumbered by the enemy forces, causing agents to favor *move*. However, as training progresses, rewards obtained from successful hit actions accumulate, the value of *hit* gradually increases, and *hit* becomes the optimal action in $Q^*$. This example concretely demonstrates the changes in the value function in realistic environments such as SMAC.
>
> Following the request of reviewer nTYN, we further analyzed value-function shifts in both SMAC-Hard (Mixture Opponent) and GRF. We kindly refer the reviewer to our detailed responses to reviewer nTYN, where these additional analyses are fully presented.
>
> ---
>
> **Question 2 (Learning dynamics of `bane_vs_hM`):**
> As the reviewer noted, the `bane_vs_hM` scenario contains substantial randomness in the initial positions, separating easy and hard cases. When Banelings start close to each other, the scenario is relatively easy, but when they spawn apart, it becomes much more difficult. Early in training, the agents quickly learn to win the easy cases, while the hard cases remain unsolved, keeping the overall win rate low. Once the agents begin to handle the harder configurations, learning accelerates sharply, resulting in rapid performance increase around 1M timesteps. We observe similar patterns in other SMAC-Comm environments, although the timing may vary across scenarios.
>
> ---
>
> **Question 4 (Relation to NA2Q [1]):**
> While both NA2Q and S2Q operate in cooperative MARL and employ multiple Q-functions, their objectives fundamentally differ. NA2Q aggregates optimal actions derived from Q-learning variants and applies an attention mechanism to obtain an interpretable value decomposition. In contrast, S2Q explicitly learns sub-value functions that sequentially capture true suboptimal joint actions derived from $Q^*$, as described in our paper. The purpose is not to diversify Q-functions for population-level benefits, but to retain and exploit precise suboptimal modes of the global value landscape. Therefore, NA2Q and S2Q pursue distinct goals and are not directly comparable in terms of design or intended functionality.
>
> ---
>
> In summary, our updates demonstrate the scalability of S2Q, while improving the clarity of its motivation and framework. We are grateful for the reviewer’s insightful comments, which allowed us to clarify these points and enhance the presentation of our results.

---

### Official Review · Reviewer_3XJX · 2025-11-02

**Soundness:** 2
**Presentation:** 2
**Contribution:** 3
**Rating:** 4
**Confidence:** 4

**Summary:**

The paper proposes Successive Sub-value Q-learning (S2Q) for cooperative MARL under CTDE. Instead of committing to a single estimated-optimal joint action, S2Q learns a sequence of monotonic, value-decomposition subnetworks that each focuses on a different high-value mode by suppressing actions selected by earlier subnetworks in its TD objective. To synchronize the index k during training, the authors use an encoder–decoder that reconstructs the global state and approximates the Softmax distribution. Experiments on SMAC Hard+, GRF, SMAC-Comm, and SMACv2 report higher win rates and faster convergence than strong baselines. Ablations suggest the Softmax selection and successive learning are key. Compute overhead is reported as modest while reaching 50% win rate substantially faster than QMIX.

**Strengths:**

1. The toy matrix-game illustrates why retaining information about nearby high-value actions can help when the optimum shifts. S2Q operationalizes this via a suppressed TD objective and Softmax-guided behavior policy. The algorithmic presentation (Alg. 1; eqs. (B.2–B.5)) is easy to follow.

2. Results span SMAC Hard+, GRF, SMAC-Comm (with a “-Comm” variant), and SMACv2, showing consistent gains and faster learning, not only final win rates. The compute table quantifies overhead.

3. Removing Softmax selection or randomizing $k$ materially hurts performance. Analysis of $K$ and temperature $T$ is informative. Sensitivity of suppression $\alpha$ and weighting $w_c$ appears in the appendix.

4. Default evaluation needs no communication relying on $Q_0^{sub}$, which is attractive compared to methods that require message passing at inference.

**Weaknesses:**

1. The paper claims theoretical/empirical analyses, but no formal result is provided to justify that minimizing the modified TD with the suppression term reliably extracts distinct top-k modes under the IGM constraint or preserves contraction/stability properties. A small lemma would strengthen the case.

2. S2Q learns an encoder–decoder to approximate $P_t$ and reconstruct $s$, which provides additional supervision such as cross-entropy and reconstruction. Several non-communication baselines do not leverage comparable auxiliary signals, raising comparability questions. A variant without the encoder–decoder, or with matched auxiliaries for baselines would calibrate the gain attributable to successive sub-values and auxiliary training.

3. While $K$ and $T$ are studied, they’re analyzed mainly on 6h vs 8z. It would be helpful to report these sweeps across more tasks (e.g., MMM2, GRF academy 4v3) and include a “no-comm” ablation that samples $k$ independently to show the necessity of synchronization, and an oracle $P_t$ using the true Softmax to bound performance.

4. In Eq. (2)/App. B.2, the target for previously selected actions is reduced by $\alpha Q^*$, which can change the scale and sign of TD targets. There is no analysis of potential instability or bias this introduces, particularly with function approximation. Clarifying normalization or clipping strategies, and why the scheme does not collapse would help. In addition, this method shares the similar idea with [1], it would be better to compare with this baseline.

[1] Wan, L., Liu, Z., Chen, X., Wang, H., & Lan, X. (2021). Greedy-based value representation for optimal coordination in multi-agent reinforcement learning. arXiv preprint arXiv:2112.04454.

**Questions:**

1. Can you formalize guarantees (even under simplified assumptions) that successive suppression + weighting (B.5) yields distinct top-k action sets, or at least that $Q_0^{sub}$ is not harmed by the auxiliary subnetworks? A proposition about bias introduced by the suppression term would be valuable.

2. What happens if you remove communication during training and have agents sample k independently, or conversely provide baselines with a matched auxiliary (state reconstruction)? Please include a “no-comm” S2Q variant and, if possible, an oracle $P_t$ variant to bound gains.

3. You show $K=2,T=0.1$ works broadly (Tab. E.2). Could you include per-map sweeps (or at least MMM2 and academy 4v3) to show consistency and to better justify the choice of $K$?

4. Since the paper cites “MARL as sequence modeling”, could you comment on (or include) a representative sequence-model baseline and discuss compatibility?

---

> ### Author Response · Authors · 2025-11-20
> **Response to Reviewer 3XJX (1/2)**
>
> ## Reviewer 3XJX
> To thoroughly address the reviewer's comments and concerns, we have divided our rebuttal into two separate comments for clarity. We kindly ask for your understanding. In the first comment, we address the theoretical analysis of successive Q-learning and the effect of auxiliary signals in non-communication settings. In the second comment, we focus on additional ablations for S2Q, its relation to the suggested baseline GVR [1], and connections to sequence modeling approaches. We sincerely appreciate your understanding and consideration.
>
> ---
>
> **Weakness 1/Question 1 (Theoretical analysis of Successive Q-learning):** We appreciate the reviewer’s insightful comments regarding the need for stronger theoretical support. To address this, we have added Theorem 4.1 in Section 4.2, together with a full proof. The theorem establishes that, under bounded rewards and with a sufficiently large suppression factor $\alpha$, each sub-value function $Q^{\mathrm{sub}}_k$ converges to the $k$-th successive suboptimal joint action of the underlying value function $Q^\ast$. In other words, minimizing the **S2Q objective provably extracts the top-$k$ modes of $Q^\ast$ in a sequential and well-defined manner.** Consequently, in S2Q, $Q_0^{\mathrm{sub}}$ is not harmed by the auxiliary sub-value networks theoretically, since each sub-value function is guaranteed to converge to its designated level in the hierarchy, maintaining a consistent and ordered structure as $K$ grows.
>
> To ensure full rigor in the proof, we slightly modified the suppression term in Equation (2), replacing $Q_{\mathrm{targ}}^\ast$ with $\max\big(Q_{\mathrm{targ}}^\ast, C\big)$, where $C>0$ is a constant introduced to properly handle potentially negative values of $Q^\ast$. This modification is used only for the theoretical analysis. In all environments considered in our experiments, $Q^\ast$ quickly becomes positive during training, so the implementation remains unchanged and directly uses $Q^\ast$ without any additional modifications. We believe this theoretical development substantially strengthens the foundation and clarity of our method. We sincerely thank the reviewer for encouraging us to include this analysis.
>
> ---
>
> **Weakness 2 (Calibrating gains from auxiliary training):** We fully understand the reviewer’s concerns regarding the potential influence of auxiliary training signals. We first clarify the role of the encoder–decoder module within S2Q. As detailed in Section 4.3, this module is used only during training to synchronize the selection of sub-value functions through the softmax distribution. Importantly, the latent representation is never fed into the $Q$-network, does not affect TD updates, and is **not used at test time**. Therefore, our primary comparison against non-communication baselines is appropriate, since S2Q does not incorporate any auxiliary features into the learned value functions during either training or evaluation.
>
> To further address the reviewer’s question regarding a version of S2Q that does inject the encoder–decoder latent directly into the utility network, such a variant corresponds to S2Q-Comm, introduced in Section 4.3. Following the reviewer’s suggestion, we also equipped non-communication baselines with the same encoder–decoder architecture and compared them against S2Q-Comm in the `SMAC-Hard+` and `GRF` tasks. The results, reported in Fig. H.3 in Appendix H.3, show that providing these latent-based auxiliary signals has minimal influence on performance in the considered environments, where communication is generally less critical. In some cases, auxiliary latents even degraded performance. Crucially, S2Q-Comm still outperforms all baselines, confirming that its gains arise from the successive sub-value learning mechanism, not from auxiliary state reconstruction or latent information.

---

> ### Author Response · Authors · 2025-11-20
> **Response to Reviewer 3XJX (2/2)**
>
> **Weakness 3 / Question 2, 3 (Additional ablation studies):**
>
> In response to the reviewer’s request, we extended the ablation studies on the hyperparameters $K$ and $T$ beyond the `6h_vs_8z` scenario to additional tasks, including `MMM2` and `academy_4_vs_3`. The results are provided in Fig. H.1 of Appendix H.1. These experiments show patterns consistent with our main findings: moderate values of $K$ such as $K=2$ offer the best balance between stability and diversity, whereas $K=0$ fails to capture meaningful suboptimal behaviors and $K=3$ introduces excessive variance. Similarly, a temperature of $T=0.1$ provides the most effective balance between convergence speed and final performance, while extreme temperatures either over-constrain or overly diffuse exploration. These observations indicate that S2Q performs reliably across tasks and that the recommended hyperparameter ranges transfer well to different environments.
>
> To further evaluate the necessity of synchronized sub-value selection, we added two additional variants to the component study in Table 1: `S2Q_oracle`, which uses the true softmax distribution $\mathbf{P}_t$ computed directly from $Q^*$, and `S2Q_independent`, where agents independently reconstruct softmax probabilities from their own observation histories without communication. As expected, `S2Q_oracle` achieves superior performance because it eliminates approximation error in estimating $\mathbf{P}_t$. In contrast, `S2Q_independent` shows a substantial performance drop, since **independently inferred probabilities cannot reliably identify the appropriate sub-value function** and fail to maintain synchronization across agents. These findings highlight that accurate softmax-guided sub-value selection and inter-agent synchronization are essential for effective exploration, further clarifying the contribution of our method. We sincerely thank the reviewer for suggesting this insightful analysis.
>
> ---
>
> **Weakness 4 (Analysis on $\alpha$):** As established in Theorem 4.1, if $\alpha$ is sufficiently large under the assumption that only actions in $\mathcal{A}\_{k-1}$ are accurately reduced, the next suboptimal action can be correctly extracted.. In practice, however, overly large values cause excessive suppression of Q-values in $\mathcal{A}_{k-1}$, and this reduction propagates to neighboring actions through function approximation, leading the learner to select actions that deviate from the true next-best mode. Conversely, when $\alpha$ is too small, suppression is insufficient and the learner struggles to move beyond previously selected actions. Our analysis in Fig. H.2(a) of Appendix H.2, conducted on the `6h_vs_8z` task for $\alpha \in \\{0.1, 0.5, 1.0, 1.5, 2.0\\}$, shows that $\alpha = 1.0$ provides the most stable results, while both $\alpha = 0.1$ and $\alpha = 2.0$ lead to noticeable performance degradation. We therefore adopt $\alpha = 1.0$ as a balanced and practically effective choice.
>
> ---
>
> **(Comparison with GVR [1]):** While we appreciate the reviewer’s suggestion to compare with GVR, the official implementation is not publicly available, which prevents a direct experimental comparison. Conceptually, the objectives and mechanisms of GVR and S2Q differ fundamentally. GVR aims to align individual greedy actions with the maximal joint Q-value by selectively replaying experiences to eliminate non-optimal self-transition nodes, reinforcing the current optimal mode. In contrast, S2Q sequentially learns sub-value functions that explicitly retain and track successive suboptimal joint actions, allowing the learner to adapt when the value landscape shifts. In other words, GVR removes suboptimal behaviors to strengthen greedy consistency, whereas S2Q preserves suboptimal behaviors as strategic alternatives for improving exploration and enabling rapid policy adaptation. These methods, therefore, pursue distinct objectives and are not directly comparable in design or purpose.
>
> ---
>
> **Question 4 (S2Q in sequence modelling):** While sequence modeling approaches in MARL often sample policies stochastically, they generally do not account for the synchronized selection across agents. Directly integrating our successive sub-value learning framework into such sequence-model architectures is nontrivial, as these models typically do not employ a QMIX-style factorization. However, we hypothesize that if suboptimal actions could be explicitly estimated according to their value and stored in a synchronized manner, similar to S2Q, this would likely improve execution fidelity and coordination among agents. In other words, leveraging value-based suboptimal action awareness with synchronized execution could be a complementary enhancement for sequence-model-based MARL.
>
> ---
>
> We thank the reviewer for their careful evaluation and constructive feedback. We trust that our clarifications and accompanying analyses directly address the concerns raised and contribute to a clearer understanding of our method.

---

### Official Review · Reviewer_nTYN · 2025-11-04

**Soundness:** 3
**Presentation:** 3
**Contribution:** 3
**Rating:** 6
**Confidence:** 3

**Summary:**

This paper proposes Successive Sub-value Q-learning (S2Q), a framework that successively learns multiple subvalue functions to retain information about alternative high-value actions. By incorporating these sub-value functions into a Softmax-based behavior policy, S2Q encourages persistent exploration and enables Qtot to adjust quickly when the optimal action changes. Experimental results show that S2Q outperforms other recent MARL methods on the StarCraft II Multi-Agent Challenges and Google Research Football.

**Strengths:**

- Dynamic value functions

    S2Q overcomes the limitation that conventional methods do not explicitly track suboptimal actions. When the optimal action changes, S2Q can immediately leverage the corresponding sub-value function and guide Q^{tot} to adapt.

- Introducing communication during training

    S2Q explicitly executes tracked suboptimal actions with priority determined by a Softmax distribution P_t over their Q^{∗} values, thereby enabling exploration of a wider range of spaces than conventional ϵ-greedy exploration.

**Weaknesses:**

- Old Benchmarks

    The StarCraft Multi-Agent Challenge (SMAC) (Samvelyan et al., 2019) is an old benchmark. It is advised to report the experimental results on the recently proposed SMAC-Hard benchmark [1].

    [1] SMAC-Hard: Enabling Mixed Opponent Strategy Script and Self-play on SMAC, arXiv:2412.17707.

**Questions:**

Figure 1 demonstrates changes in the payoff matrix. Could you provide concrete examples of how the value function dynamically evolves in SMAC-Hard or GRF environments?

---

> ### Author Response · Authors · 2025-11-20
> **Response to Reviewer nTYN**
>
> ## Reviewer nTYN
> We sincerely thank the reviewer for their insightful comments and suggestions. In particular, regarding evaluation on recent SMAC-Hard benchmarks, we provide additional performance comparisons on these benchmarks, along with detailed examples illustrating how the value function evolves during training.
>
> ---
>
> **Weakness 1 (Evaluation in recent benchmarks):** Thank you for your suggestion. As discussed in Lines 397–398, in addition to our evaluation on the original SMAC environments, we have also conducted experiments on SMACv2 [R.1], a recently proposed benchmark that incorporates additional stochasticity through randomized team compositions and initial positions. The performance comparison in SMACv2 tasks is provided in Appendix E.4. As shown in Fig. G.1, S2Q consistently outperforms other MARL baselines across all considered scenarios: `terran_5_vs_5`, `zerg_5_vs_5`, and `protoss_5_vs_5`. These results demonstrate that S2Q is effective not only in the standard SMAC setting but also in newer, more stochastic, and more realistic benchmarks.
>
> Furthermore, following the reviewer’s recommendation, we evaluated S2Q on the recently proposed SMAC-Hard benchmark, where opponents adopt mixed scripted strategies that introduce higher levels of stochasticity and non-stationarity. To distinguish it from the original SMAC environment, we append `_Hard` to each scenario name, including `5m_vs_6m_Hard`, `2s3z_Hard`, and `3s_vs_5z_Hard`. The results, provided in Appendix G.2 and summarized in Fig. G.2, show that **S2Q again achieves consistently superior performance over all baselines**. These findings confirm that S2Q maintains strong and stable performance even in newly proposed, highly stochastic MARL environments.
>
> [R.1] Ellis, Benjamin, et al. ``Smacv2: An improved benchmark for cooperative multi-agent reinforcement learning." *Neurips 2023*
>
> ---
>
> **Question 1 (Change in value function):** Thank you for your question. In the SMAC environment, we illustrated how the value function evolves during training using the `6h_vs_8z` scenario in Fig. 6. Early in training, the learned value function assigns a higher value to evasive behaviors, as the ally agents are outnumbered by the enemy forces, causing agents to favor *move*. However, as rewards obtained from successful hit actions accumulate, the value landscape changes: as the value of *hit* gradually increases, it becomes the optimal action in $Q^\ast$ as training progresses. S2Q adapts by shifting $Q_0^{\mathrm{sub}}$ to prioritize *hit* while still leveraging $Q_1^{\mathrm{sub}}$ and $Q_2^{\mathrm{sub}}$ to track *move*, the previously optimal behavior. This example concretely demonstrates the changes in the value function in realistic environments such as SMAC, and how S2Q follows and responds.
>
> To further clarify this phenomenon, we additionally examined value-function shifts in `5m_vs_6m_Hard` of SMAC-Hard and `academy_4_vs_3` of GRF, and added the results in Fig. H.4 and H.5 of Appendix H.4. In `5m_vs_6m_Hard`, agents initially attack greedily to secure easy eliminations, but the value function later learns that repositioning to form a favorable formation yields higher long-term returns, causing the optimal action to shift from *attack* to *move*. A similar trend appears in `academy_4_vs_3`, where agents first take low-success long shots but eventually learn that approaching the goal or passing increases scoring probability, shifting the optimal action from *shoot* to *pass*. These results further illustrate how S2Q tracks and adapts to evolving optimal behaviors across environments.
>
> ---
>
> Overall, the additional experiments and analyses demonstrate that S2Q effectively adapts to evolving value landscapes and maintains consistent performance in challenging and stochastic multi-agent environments. We sincerely thank the reviewer for their valuable feedback, which helped us clarify these aspects and strengthen our empirical results.

---

### Author Response · Authors · 2025-11-20
**General response to all reviewers**

## General Response to All Reviewers
We sincerely thank all reviewers for their constructive feedback. Following your suggestions, we have substantially strengthened the manuscript with additional proof and analyses to better demonstrate our framework. A revised version, with all changes highlighted in blue, has been uploaded. The major updates are summarized below.

---

**(i) Theoretical guarantee of successive Q-learning (Section 4.2, Appendix B):** Following requests for stronger theoretical support, we added Theorem 4.1 together with a full proof. The theorem formally establishes that, under bounded rewards and a sufficiently large suppression factor $\alpha$, each sub-value function $Q^{\mathrm{sub}}_k$ converges to the $k$-th successive suboptimal joint action of the underlying value function. This demonstrates that minimizing the **S2Q objective provably extracts the top-$k$ modes of $Q^*$ in a sequential and well-defined manner.**

**(ii) Broader environments and additional ablations (Section 5.1, 5.3, Appendix G.2, H.1, H.2, H.3, H.5):** To evaluate S2Q for its robustness under more challenging and diverse conditions, we have included experiments in the SMAC-Hard (Mixture Opponent) benchmark, where S2Q showed consistent improvements over baselines. We also provide a deeper analysis of the hyperparameters of S2Q, along with a more in-depth computational cost study. These additions provide broader empirical support and clarify S2Q’s scalability.

**(iii) Clearer motivation and expanded related work (Section 5.2, Appendix H.4, Section 3.2):** We improved clarity in the main text by providing additional concrete examples of the shift in the value function from SMAC-Hard (Mixture Opponent) and Google Research Football, further highlighting the key motivation of S2Q. Meanwhile, we expand our related works within existing MARL research to reflect reviewer suggestions.

---

We believe these revisions address the main concerns raised during review and improve the clarity and completeness of the paper. Most notably, the addition of the theoretical guarantee for successive Q-learning provides a rigorous foundation for S2Q, formally ensuring that each sub-value function reliably captures the top-$k$ modes of $Q^*$ in a sequential and well-defined manner. We are grateful for the reviewers’ guidance, which materially enhanced the manuscript.

---

### Comment · Area_Chair_Ncnf · 2025-11-24

Dear Reviewers,

The authors have responded to your reviews. Please review and respond to their comments who have not yet done so.

Best, Your AC

---

### Author Response · Authors · 2025-12-02
**Response to Area Chair**

Dear Area Chair,

Thank you very much for your time and effort in handling our submission. In addition to the reviewer-specific replies and common responses provided below, we briefly summarize the main contribution of our work and how we addressed the reviewers’ concerns, so that the revisions are easier to follow from your perspective as Area Chair.

---

**Main contribution:** The proposed S2Q algorithm targets a fundamental limitation of standard value-based MARL methods that rely on IGM, which focus exclusively on accurately estimating the value-maximizing joint action. Due to poor representation of suboptimal actions, these methods depend on inefficient $\epsilon$-greedy exploration, which makes the policy slow to adapt when values change. S2Q addresses this limitation by **learning sub-value functions that assign accurate values to each suboptimal action**, allowing softmax-style exploration instead of relying on $\epsilon$-greedy. Through a payoff-game formulation, we show that this design enhances both responsiveness and adaptability to value changes, and our experiments across diverse environments demonstrate that S2Q achieves significantly higher performance and efficiency compared to other existing methods.

---

**Summary of rebuttal:** The reviewers’ comments suggest that all reviewers recognized S2Q's distinct contribution and strong empirical performance. Their main requests focused on three aspects: the need for theoretical analysis (reviewers *3XJX* and *44v8*), additional baseline comparisons (reviewers *nTYN* and *44v8*), and more extensive ablation studies (all reviewers), including value change analysis and scalability with respect to $K$. In response, we added a new **Theorem 4.1, which provides a theoretical guarantee that S2Q can correctly identify and utilize suboptimal actions** and thus reinforces the core claim of the paper; this was one of the most important improvements made during the rebuttal period. We also added new experiments in SMAC-Hard (mixture-opponent) settings and carried out all suggested ablations, showing that S2Q continues to outperform other baselines and adapts well to value changes across a wider range of scenarios. We believe these additions address the reviewers’ concerns in detail and significantly improve the paper.

---

It is unfortunate that we could not receive follow-up comments from most reviewers due to an OpenReview system issue. However, **reviewer *44v8*, who initially raised the largest number of concerns, stated that most of their concerns had been resolved and accordingly increased their score to 6**, as reflected in the rebuttal thread. Although other reviewers were not able to submit follow-up comments, reviewer *3XJX* raised concerns that were very similar in nature to those of reviewer *44v8*, aside from a few additional analysis requests. Therefore, we believe their opinion would likely have evolved in a direction similar to that of reviewer *44v8*. The remaining reviewers, reviewer *nTYN* and reviewer *xg2N*, had already expressed positive views in their initial reviews and raised fewer concerns compared to the other two reviewers, mainly requesting additional experiments and analysis, which we were able to conduct and describe in detail in the revised version and rebuttal. For these reasons, we expect that their overall assessments would have remained positive, in line with their original evaluations.

In summary, we believe that S2Q offers clear novelty and a distinct contribution. During the rebuttal period, the central theoretical concerns were substantially resolved, and the extended experiments and ablations have further strengthened the paper and addressed the main points raised by the reviewers. We hope this overview is helpful for your decision, and we are sincerely grateful for your careful consideration of our work.

---

### Meta-Review · Area_Chair_A1Wh · 2026-01-02

**Summary:**

The submission received mixed reviews. Before the rebuttal, the scores are 4468. The main concerns about the submissions can be summarized as follows:
1. The benchmark SMAC is relatively old; more comprehensive evaluations are needed.
2. The theoretical depth is limited; there are no formal results to justify the proposed method.
3. The comparisons with prior works may be unfair.
4. The selection standard of hyperparameters is not clear.
5. The discussions with some mentioned related works are necessary, such as the high-order Q-learning method.
6. A quantitative analysis comparing GPU memory is needed.
7. Some experimental details are missing.

After the rebuttal, most of the concerns have been addressed in the revision. The only arguable concern is the "theoretical depth" (although the author provided Theorem 4.1 in the submission).

Overall, I think the submission is a good submission with extensive empirical results to support all claims.

**Reviewer Concerns:**

As mentioned, all concerns about more comprehensive evaluations are well-addressed, as the authors have provided many new experiments in the revision. The remaining concerns are: 1) a deeper theoretical analysis about the proposed method; 2) some more realistic examples abou the shift scenarios beyond the toy example.

**Reviewer Scores:**

I think the reviewer nTYN and reviewer xg2N would like to maintain their original rating as 6, 8. Both two "negative" reviewers (3XJX and 44v8) would like to increase their ratings to 6.

---

### Decision · Program_Chairs · 2026-01-26

Accept (Poster)